# Metabolic flux from the Krebs cycle to glutamate transmission tunes a neural brake on seizure onset

**Jiwon Jeong, Jongbin Lee, Ji-hyung Kim, Chunghun Lim** *

Department of Biological Sciences, Ulsan National Institute of Science and Technology, Ulsan, Republic of Korea

* clim@unist.ac.kr

**Data Availability Statement:** All relevant data are within the manuscript and its Supporting Information files.

**Funding:** This work was supported by grants from the Suh Kyungbae Foundation (SUHF-17020101

## Abstract

Kohlschütter-Tönz syndrome (KTS) manifests as neurological dysfunctions, including early-onset seizures. Mutations in the citrate transporter *SLC13A5* are associated with KTS, yet their underlying mechanisms remain elusive. Here, we report that a *Drosophila SLC13A5* homolog, *I'm not dead yet* (*Indy*), constitutes a neurometabolic pathway that suppresses seizure. Loss of *Indy* function in glutamatergic neurons caused "bang-induced" seizure-like behaviors. In fact, glutamate biosynthesis from the citric acid cycle was limiting in *Indy* mutants for seizure-suppressing glutamate transmission. Oral administration of the rate-limiting α-ketoglutarate in the metabolic pathway rescued low glutamate levels in *Indy* mutants and ameliorated their seizure-like behaviors. This metabolic control of the seizure susceptibility was mapped to a pair of glutamatergic neurons, reversible by optogenetic controls of their activity, and further relayed onto fan-shaped body neurons via the ionotropic glutamate receptors. Accordingly, our findings reveal a micro-circuit that links neural metabolism to seizure, providing important clues to KTS-associated neurodevelopmental deficits.

## Author summary

Kohlschütter-Tönz syndrome (KTS) is a neurodevelopmental disorder linked to two distinct genomic loci encoding the citrate transporter SLC13A5 and synaptic protein ROGDI, respectively. An early-onset seizure is the most prominent neurological symptom in KTS patients, yet how these genes contribute to the control of seizure susceptibility remains poorly understood. Our study establishes behavioral models of seizure in *Drosophila* mutants of KTS-associated genes and demonstrates a genetic, metabolic, and neural pathway of seizure suppression. We discover that the metabolic flux of the Krebs cycle to glutamate biosynthesis plays a critical role in scaling seizure-relevant glutamate transmission. We further map this seizure-suppressing pathway to a surprisingly small number of glutamatergic neurons and their ionotropic glutamate transmission onto a key sleep-promoting locus in the adult fly brain. Given that the excitatory amino acid glutamate is considered a general seizure-promoting neurotransmitter, our findings illustrate how glutamatergic transmission can have opposing effects on seizure susceptibility in the context

[CL]); from the National Research Foundation funded by the Ministry of Science and Information & Communication Technology (MSIT), Republic of Korea (NRF-2018R1A5A1024261[CL]; NRF-2021M3A9G8022960[CL]), and by the Ministry of Education, Republic of Korea (NRF-2019R1I1A1A01063087[JL]). The funders had no role in study design, data collection and analysis, decision to publish, or preparation of the manuscript.

**Competing interests:** The authors have declared that no competing interests exist.

of a micro-neural circuit, possibly explaining drug-resistant epilepsy. This seizure-suppressing locus in the *Drosophila* brain is also implicated in metabolism, circadian rhythms, and sleep, revealing the conserved neural principles of their intimate interaction with epilepsy across species.

## Introduction

Kohlschütter-Tönz syndrome (KTS) is a genetic disorder that manifests as developmental abnormalities such as tooth dysplasia, intellectual disability, and early-onset epileptic encephalopathy [1]. Whole-exome sequencing has revealed molecular lesions that are associated with KTS, identifying mutations in two independent loci, *ROGDI* [2–6] and *SLC13A5* (solute carrier family 13 member 5) [7–10], as the genetic causes of KTS. Homologs of the two KTS-associated genes are relatively well conserved across different species. Accordingly, animal models for *ROGDI* or *SLC13A5* serve as essential genetic resources to elucidate the physiological function of the gene products and their mechanisms underlying the KTS pathogenesis.

SLC13A5 is a plasma membrane transporter of intermediates of the tricarboxylic acid (TCA) cycle, also known as the citric acid cycle or Krebs cycle, that displays the highest affinity for citrate [11,12]. The *Drosophila* homolog of *SLC13A5* is *I'm not dead yet* (*Indy*), named for the long lifespan of loss-of-function mutants [13], although the longevity phenotype might be sensitive to genetic background or calorie intake [14–16]. As expected based on its citrate-transporter activity, loss of *Indy* function phenocopies physiological changes in calorie-restricted animals, including low body weight, low triglyceride levels, and high sensitivity to starvation [16,17]. In fact, calorie restriction down-regulates *Indy* expression in *Drosophila*, whereas *Indy* mutants display low expression levels of insulin-like peptides [16]. Mitochondrial physiology is also altered in *Indy* mutant flies (i.e., increased mitochondrial biogenesis, decreased electron transport chain function), likely via a mechanism dependent on peroxisome proliferator-activated receptor gamma coactivator-1α [16–18].

Consistent with this, depletion of an INDY homolog in worms extends life span and leads to a "lean" phenotype, likely via an AMPK-dependent pathway [19,20]. In mice, genomic deletion of the mammalian *Indy* homolog (*mIndy*), as well as liver-specific mINDY depletion, results in a protection against obesity, fatty liver, and insulin resistance upon feeding a high-fat diet—metabolic conditions comparable to those observed in flies and worms [21–23]. Notably, gene expression profiles in *mIndy* mutant mice largely resemble those in calorie-restricted animals [21]. *mIndy* expression in mice is also altered by metabolic challenge (e.g., starvation, fatty-liver disease conditions), and distinct transcription factors (e.g., CREB, STAT3, ARNT, PXR) have been implicated in this adaptive regulation of *mIndy* transcription [24–27].

While there is abundant genetic evidence for *Indy* function in metabolism [28–30], much less is known about the function of *ROGDI* homologs. A *Drosophila* genetic screen initially identified *rogdi* as one of the genes associated with memory formation [31] and emerging evidence suggests a role of *ROGDI* homologs in neurons [3,5,32,33]. The crystal structure of human ROGDI protein displays a leucine-zipper-like four-helix bundle and a characteristic beta-sheet domain, hinting at how KTS-associated *ROGDI* mutations would impair the overall structure and stability of the encoded protein [34]. Nonetheless, how metabolic or neural functions of the KTS-associated gene products are coupled to the neurodevelopmental pathogenesis underlying KTS remains elusive.

In this study, we demonstrate that *Drosophila* mutants of KTS-associated genes display seizure-like behaviors. We further provide compelling evidence that *Indy* links the flux of the

TCA cycle in a specific set of glutamatergic neurons to the scale of their neural transmission to achieve metabolic control of seizure susceptibility. Given that early-onset seizure is one of the most prominent symptoms in KTS patients, our findings provide important insights into the neural mechanism responsible for the development of KTS.

## Results

### *Indy* acts in glutamatergic neurons to suppress bang-induced seizure

The early-onset seizure is one of the most prominent neurological symptoms observed in KTS patients. We hypothesized that *Drosophila* mutants of KTS-associated genes might recapitulate genetic conditions in KTS patients and display seizure-like behaviors. After a mechanical stimulus (vortexing for 25 s), wild-type flies immediately recovered a normal posture and resumed their locomotion (Fig 1A). By contrast, *Indy* mutants homozygous or trans-heterozygous for loss-of-function alleles exhibited "bang-induced" seizure-like behaviors (e.g., severe wing-flapping, abdominal contractions, leg-twitching, or failure to stand upright), as reflected in a high seizure index and prolonged recovery time (Fig 1B and S1 Movie). The bang-sensitive seizure (BSS) phenotypes in *Indy* mutants were comparable to those observed in other seizure mutants, such as *easily shocked* or *slamdance* but weaker than *bang senseless* mutants [35–37] (S1A Fig). Moreover, *Indy* mutant seizure displayed transient resistance to the second mechanical stimulus after the first BSS (S1B Fig), indicating the seizure threshold shifts during the refractory period as observed in other bang-sensitive mutants [38]. To map a neural locus responsible for *Indy* mutant BSS, we silenced *Indy* expression by overexpressing an RNA interference (RNAi) transgene in select groups of cells and examined subsequent effects on BSS (S2A Fig). INDY depletion in vesicular glutamate transporter (*VGlut*)-expressing neurons phenocopied BSS in *Indy* mutants (Fig 1C), whereas the *Indy* RNAi in other groups of neurons defined by their specific neurotransmitters (e.g., GABAergic, cholinergic, or dopaminergic neurons) did not induce BSS (S2B Fig). The INDY-depletion phenotypes were confirmed by independent *Indy* RNAi transgenes (i.e., *Indy*$^{RNAi}$ #2, #3, and #4), possibly excluding off-target effects (S2C Fig). We generated a mutant INDY transgene that harbored a KTS-associated allele (S3 Fig, INDY$^{T245M}$) and expressed mutant INDY proteins with no citrate transporter activity [7–9,39]. Overexpression of INDY$^{T245M}$ in *VGlut*-expressing neurons similarly induced BSS in a wild-type *Indy* background (Fig 1C). Moreover, transgenic expression of wild-type *Indy* cDNA in *VGlut*-expressing neurons was sufficient to rescue BSS in *Indy* mutants (Fig 1D). These results suggest that *Indy* function in glutamatergic neurons is necessary and sufficient for seizure suppression. We further found that *rogdi* mutants displayed BSS comparably to *Indy* mutants (S4A Fig), and its seizure-suppressor function was similarly mapped to glutamatergic neurons (S4B–S4D Fig). Nevertheless, our subsequent analyses focused on elucidating *Indy*-dependent mechanisms of seizure control since the molecular function of ROGDI has been poorly defined.

### Down-regulation of glutamate transmission induces BSS in *Indy* mutants

To determine if glutamate transmission actually contributes to BSS phenotypes in *Indy* mutants, we genetically manipulated the expression of VGLUT, a vesicular transporter that incorporates glutamate into synaptic vesicles [40], and examined subsequent effects on *Indy*-dependent BSS. The heterozygosity of *VGlut* did not induce seizure-like behaviors in wild-type flies (Fig 2A); however, it substantially increased seizure susceptibility in *Indy* heterozygous mutants. VGLUT overexpression in glutamatergic neurons partially but significantly rescued BSS phenotypes in INDY-depleted flies (Fig 2B). It has been shown that VGLUT overexpression increases synaptic vesicle size in larval motor neurons and their spontaneous release of

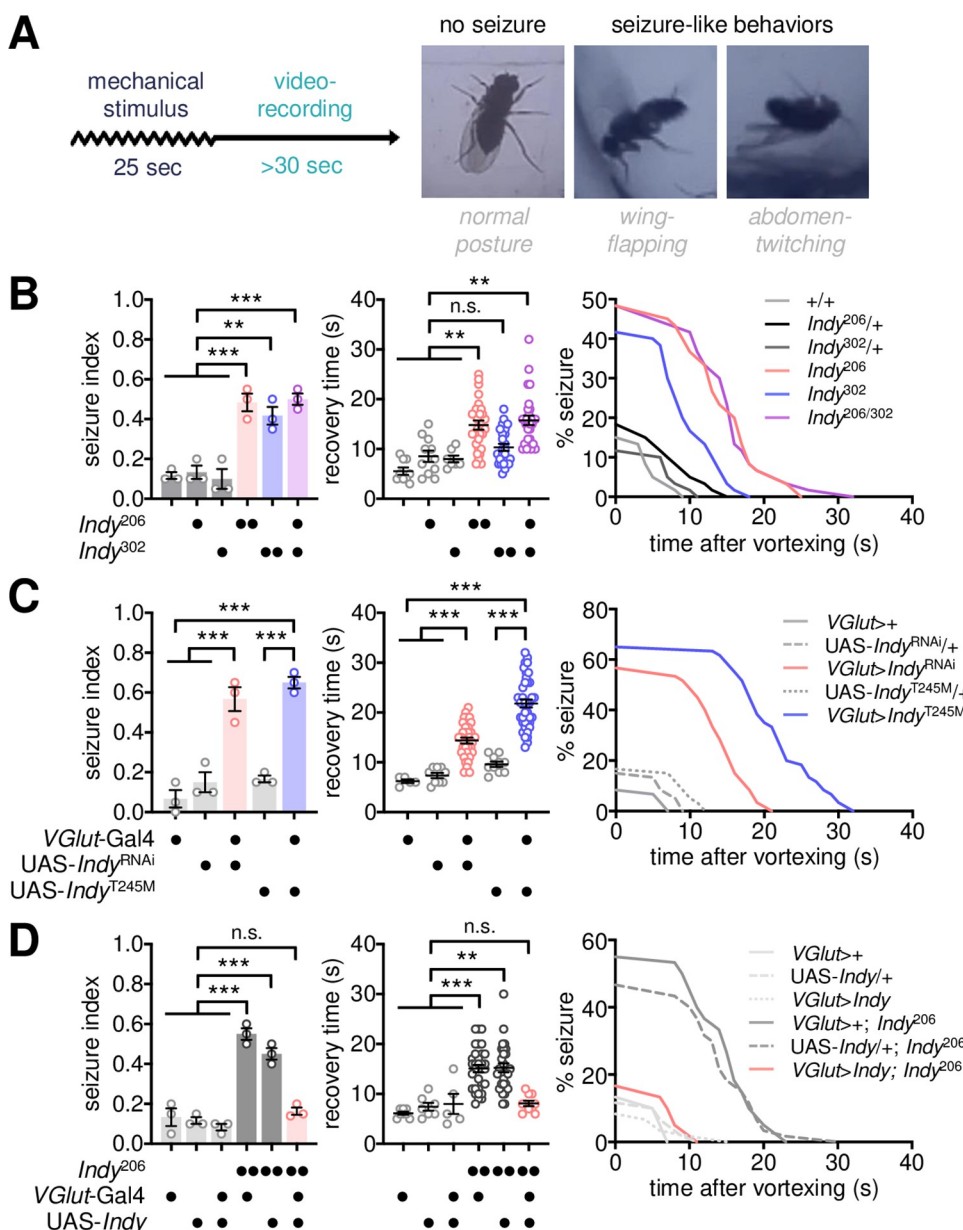

**Fig 1. Loss of *Indy* function in glutamatergic neurons induces BSS.** (A) Experimental design for the quantitative analysis of bang-induced seizure-like behaviors in *Drosophila*. A group of flies (n = 5) was vortexed for 25 s, after which post-stimulus responses were video recorded. Seizure-like behaviors, including wing-flapping and abdomen-twitching, were scored in individual flies. (B) Quantitative analyses of BSS in *Indy* mutants homozygous or trans-heterozygous for loss-of-function alleles. A seizure index was calculated as the ratio of the number of BSS-positive flies to the total number of flies tested in each experiment (n = 20; 5 flies per group × 4 groups per experiment) and averaged from three independent experiments. Recovery time was calculated individually for BSS-positive flies as the latency to normal posture after vortexing and was averaged for each genotype (n = 7–29 flies). Percent seizure was calculated as the percentage of BSS-positive flies per genotype at each second after vortexing (n = 60 flies; 20 flies per experiment × 3 experiments). Data represent means ± SEM. n.s., not significant; $^{**}P < 0.01$, $^{***}P < 0.001$, as determined by one-way ANOVA with Holm-Sidak's multiple comparisons test (seizure index) or by Kruskal Wallis test with Dunn's multiple comparisons test (recovery time). (C) Silencing of *Indy* function in glutamatergic neurons by transgenic overexpression of *Indy*^RNAi or KTS-associated *Indy*^T245M is sufficient to induce BSS. Quantitative analyses of BSS in individual flies were performed as described above. Data represent means ± SEM (seizure index, n = 60 flies in 3 independent experiments; recovery time, n = 5–39 flies). $^{***}P < 0.001$, as determined by one-way ANOVA with Holm-Sidak's multiple comparisons test. (D) Transgenic overexpression of wild-type INDY in glutamatergic neurons rescues BSS in *Indy* mutants. Data represent means ± SEM (seizure index, n = 60 flies in 3 independent experiments; recovery time, n = 5–33 flies). n.s., not significant; $^{**}P < 0.01$, $^{***}P < 0.001$, as determined by two-way ANOVA with Holm-Sidak's multiple comparisons test.

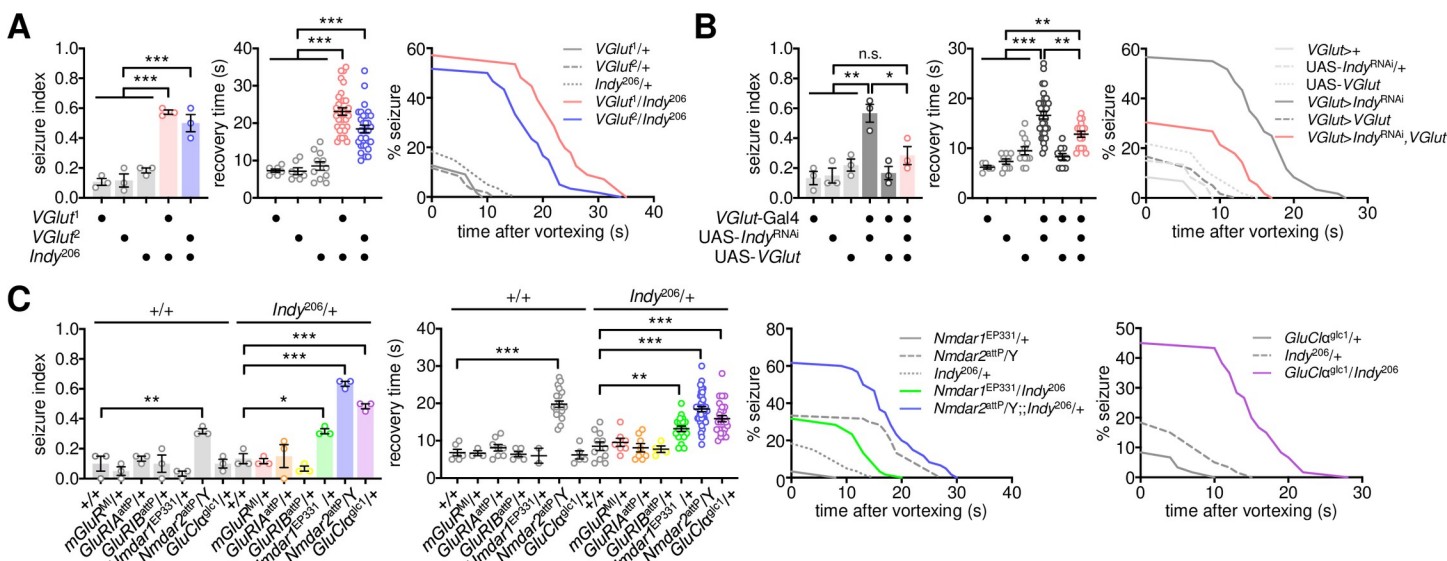

**Fig 2. Down-regulation of glutamatergic transmission is responsible for *Indy*-dependent BSS.** (A) Heterozygosity of *VGlut* induces BSS in heterozygous *Indy* mutants. Quantitative analyses of BSS in individual flies were performed as described in Fig 1. Data represent means ± SEM. ***$P < 0.001$, as determined by one-way ANOVA with Holm-Sidak's multiple comparisons test (seizure index, n = 55–60 flies in 3 independent experiments) or by Welch's ANOVA with Dunnett's multiple comparisons test (recovery time, n = 7–32 flies). (B) Overexpression of wild-type VGLUT suppresses BSS induced by transgenic depletion of INDY in glutamatergic neurons. Data represent means ± SEM (seizure index, n = 55–60 flies in 3 independent experiments; recovery time, n = 5–34 flies). n.s., not significant; *$P < 0.05$, **$P < 0.01$, ***$P < 0.001$, as determined by one-way ANOVA with Holm-Sidak's multiple comparisons test. (C) Heterozygosity of ionotropic glutamate receptor *Nmdar* or *GluClα* induces BSS in heterozygous *Indy* mutants. Significant *Indy* x glutamate receptor interaction effects on seizure index ($P = 0.0047$ for *Nmdar1*; $P = 0.0023$ for *Nmdar2*; $P = 0.0009$ for *GluClα*) and recovery time ($P = 0.0136$ for *Nmdar2*; $P = 0.0049$ for *GluClα*) were detected by two-way ANOVA. Data represent means ± SEM (seizure index, n = 60 flies in 3 independent experiments; recovery time, n = 2–37). *$P < 0.05$, **$P < 0.01$, ***$P < 0.001$, as determined by Holm-Sidak's multiple comparisons test.

glutamate at the larval neuromuscular junction [40–42], whereas it could lead to neurodegeneration in a cell type-specific manner [42,43]. Nonetheless, no gross effects of VGLUT overexpression were detected on the seizure susceptibility in wild-type flies (Fig 2B). We further mapped two glutamate receptors, N-methyl-D-aspartic acid receptors (*Nmdar1* and *Nmdar2*) and glutamate-gated chloride channel (*GluClα*), that mediate *Indy*-dependent BSS in transheterozygous mutants (Fig 2C). However, hemizygous *Nmdar2* mutants exhibited significant BSS even in a wild-type *Indy* background. These genetic interactions suggest that down-scaling of glutamate transmission via the ionotropic glutamate receptors may underlie BSS phenotypes in *Indy* mutants.

## A metabolic link between the TCA cycle and glutamate transmission underlies *Indy*-dependent seizure suppression

Two classes of presynaptic neurons—glutamatergic and GABAergic—are intrinsically coupled to astrocytes via the glutamate/GABA-glutamine cycle, mediating excitatory and inhibitory transmissions, respectively. An imbalance in their activity is associated with neurological disorders, including a seizure [44–46]. In contrast to the mammalian central nervous system, acetylcholine is the primary excitatory neurotransmitter in the fly brain whereas glutamate plays this role at the neuromuscular junction [47–49]. Nonetheless, our genetic evidence indicated that low glutamate transmission likely induced seizure-like behaviors in *Indy* mutants. We thus examined if the loss of *Indy* function impaired the biosynthesis of glutamate or GABA. Our quantitative assessment of free amino acids revealed that glutamate levels were substantially reduced in *Indy* mutants (Fig 3A). In contrast, no significant differences in GABA levels were detected between wild-type and *Indy* mutant flies. Considering that glutamate

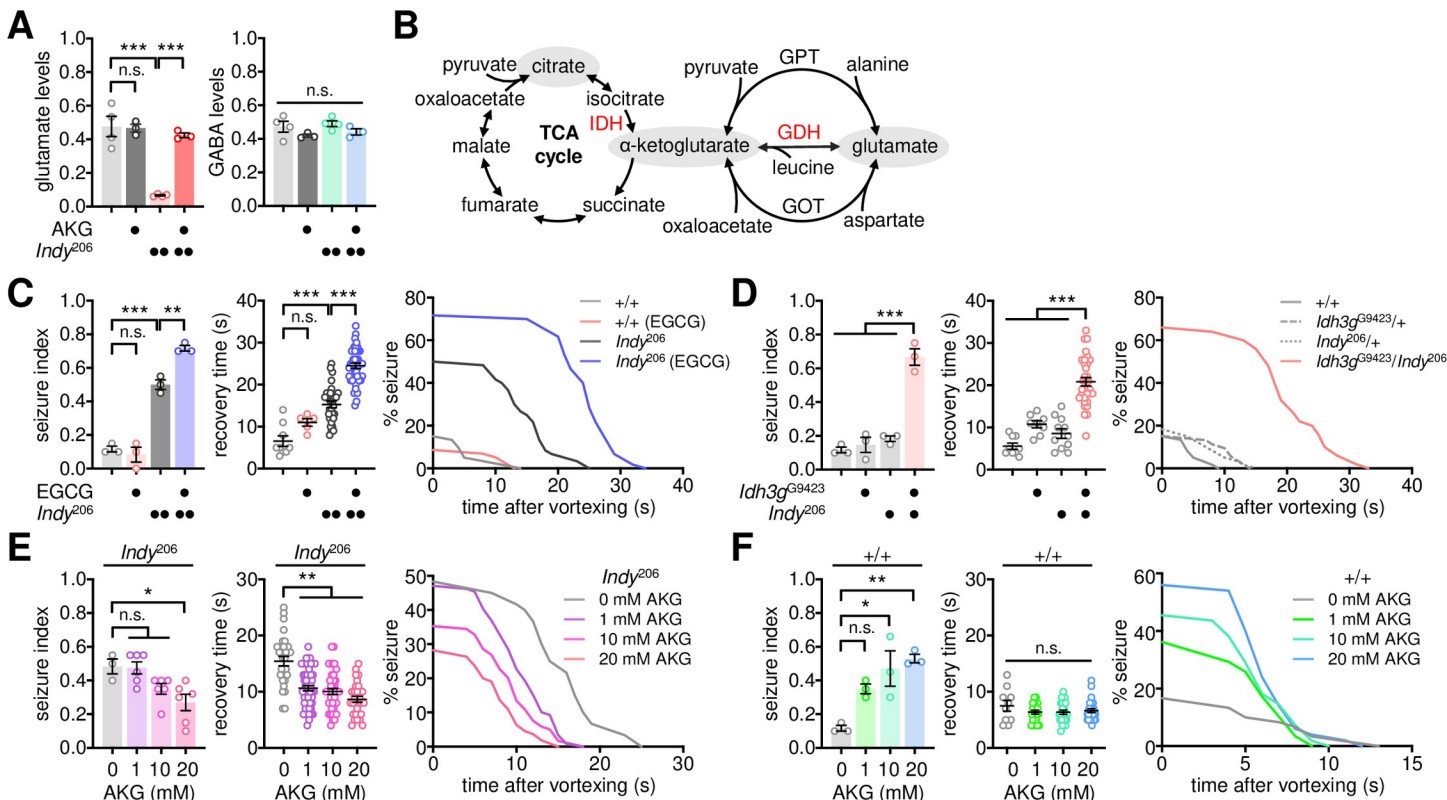

**Fig 3. *Indy*-dependent BSS involves a metabolic link between the TCA cycle and glutamate transmission.** (A) *Indy* mutants display low levels of free glutamate that are rescued by oral administration of α-ketoglutarate (AKG). Flies were fed control or AKG-containing food (20 mM) for 3 d before harvesting. Free amino acids in whole-body extracts were quantified using ion-exchange chromatography. Relative levels of glutamate and GABA were calculated by normalizing to glycine levels. A significant *Indy* x AKG interaction effect on glutamate levels ($P = 0.0006$) was detected by two-way ANOVA. Data represent means ± SEM (n = 3–4). n.s., not significant; ***$P < 0.001$, as determined by Holm-Sidak's multiple comparisons test. (B) Glutamate biosynthesis via the TCA cycle. IDH, isocitrate dehydrogenase; GDH, glutamate dehydrogenase; GPT, glutamate-pyruvate transaminase; GOT, glutamate-oxaloacetate transaminase. (C) Oral administration of the GDH inhibitor epigallocatechin gallate (EGCG) exaggerates BSS phenotypes in *Indy* mutants. Flies were fed control or EGCG-containing food (5 mg/ml) for 3 d before the BSS assessment. Quantitative analyses of BSS in individual flies were performed as described in Fig 1. Two-way ANOVA detected a significant *Indy* x EGCG interaction effect on seizure index ($P = 0.0025$). Data represent means ± SEM (seizure index, n = 58–60 flies in 3 independent experiments; recovery time, n = 5–43 flies). n.s., not significant; **$P < 0.01$, ***$P < 0.001$, as determined by Holm-Sidak's multiple comparisons test. (D) *Idh3g* heterozygosity induces BSS phenotypes in heterozygous *Indy* mutants. Significant *Indy* x *Idh3g* interaction effects on seizure index ($P = 0.0002$ by two-way ANOVA) and recovery time ($P = 0.0086$ by Aligned ranks transformation ANOVA) were detected. Data represent means ± SEM (seizure index, n = 50–60 flies in 3 independent experiments; recovery time, n = 8–33 flies). ***$P < 0.001$, as determined by Holm-Sidak's multiple comparisons test (seizure index) or by Wilcoxon rank sum test (recovery time). (E and F) Oral administration of AKG rescues BSS phenotypes in *Indy* mutants. Flies were fed control or AKG-containing food (20 mM) for 3 d before the BSS assessment. Data represent means ± SEM (seizure index, n = 50–119 flies in 3 independent experiments; recovery time, n = 10–56 flies). n.s., not significant; *$P < 0.05$, **$P < 0.01$, as determined by one-way ANOVA with Holm-Sidak's multiple comparisons test (seizure index), by Welch's ANOVA with Dunnett's T3 multiple comparisons test (E, recovery time), or by Kruskal-Wallis test with Dunn's multiple comparisons (F, recovery time).

dehydrogenase (GDH) mediates the interconversion of glutamate and α-ketoglutarate—one of the TCA cycle intermediates (Fig 3B)—we reasoned that loss of INDY-dependent import of extracellular citrate might limit glutamate biosynthesis via the TCA cycle.

To validate this hypothesis, we fed the GDH inhibitor, epigallocatechin gallate [50], to adult flies and tested its effects on BSS phenotypes. Pharmacological inhibition of GDH robustly increased both seizure index and recovery time after BSS in *Indy* mutants, but it did not induce BSS in wild-type flies (Fig 3C). Similar results were obtained using another GDH inhibitor (i.e., diethylstilbestrol), excluding possible off-target effects (S5 Fig). Accordingly, these data suggest that GDH-mediated metabolic flux is limiting for seizure suppression, particularly in the context of *Indy* deficiency. To further examine the involvement of the TCA cycle in seizure suppression, we genetically manipulated isocitrate dehydrogenase 3 (IDH3), a hetero-

tetrameric enzyme that converts isocitrate into α-ketoglutarate in the rate-limiting step in the TCA cycle (Fig 3B). Heterozygosity of the *Idh3g* mutant allele induced BSS in *Indy* heterozygous mutants, but not in wild-type flies (Fig 3D). In addition, RNAi-mediated depletion of individual IDH3 subunit proteins in wild-type glutamatergic neurons alone was sufficient to induce BSS (S6 Fig). These lines of pharmacological and genetic evidence indicate that reducing the metabolic flux from the TCA cycle to glutamate biosynthesis may down-scale seizure-suppressing glutamate transmission, thereby causing BSS phenotypes in *Indy* mutants.

We further hypothesized that dietary supplements of TCA cycle intermediates should compensate for the genetic deficits in flies with loss of *Indy* function and rescue their BSS phenotypes. Oral administration of α-ketoglutarate indeed ameliorated seizure phenotypes in *Indy* mutants in a dose-dependent manner (Fig 3E). Moreover, α-ketoglutarate supplementation restored glutamate levels in *Indy* mutants to wild-type levels (Fig 3A). Unexpectedly, we found that α-ketoglutarate supplementation induced a dose-dependent increase in seizure index, but not recovery time, in wild-type flies (Fig 3F). A possible explanation for this observation is that an excess of α-ketoglutarate may lead to an imbalance in the metabolic flux between the TCA cycle and glutamate in wild-type flies, thereby lowing the threshold for seizure initiation. Alternatively, surplus α-ketoglutarate may deplete synaptic vesicles containing glutamate by promoting their fusion with the synaptic membrane [51] (see Discussion).

## A pair of glutamatergic neurons mediates *Indy*-dependent seizure suppression

Genetic and biochemical analyses in *Indy* mutants revealed the contribution of low glutamate transmission to their seizure phenotypes. We found that daily locomotor activity was reduced in *Indy* mutants, but their waking activity (i.e., activity count per minute awake) was indistinguishable from wild-type control (S7A Fig). Moreover, wild-type and *Indy* mutants displayed similar climbing activities in a negative geotaxis assay (S7B Fig). We thus reasoned that the glutamate transmission in *Indy* mutants might not be limiting for general motor function as reported previously [13,16,52–54], but a subset of the glutamatergic neurons would be somehow sensitized to loss of *Indy* function for seizure control. Our transgenic mapping of the *Indy* RNAi phenotypes actually identified a very small group of neurons expressing the neuropeptide leucokinin [55] (hereafter, LK neurons) as a neural locus important for *Indy*-dependent seizure suppression (Figs 4A and S2B). Introduction of a temperature-sensitive Gal80 transgene allowed us to turn off the Gal4-driven expression of the *Indy* RNAi transgene at low temperature (i.e., 21˚C) but silence endogenous *Indy* expression in LK neurons at high temperature (i.e., 29˚C). The conditional RNAi confirmed that *Indy* was necessary in adult LK neurons for seizure suppression (S8 Fig). Conversely, transgenic expression of the wild-type *Indy* cDNA in LK neurons was sufficient to rescue BSS in *Indy* mutants (Fig 4B).

LK neurons can be divided into three groups based on their neuroanatomical positions in the adult brain and ventral nerve cord [56]. These include lateral horn LK (LHLK), subesophageal ganglion LK (SELK), and abdominal LK (ABLK) neurons (Fig 4A). To determine which subset of LK neurons contributed to *Indy*-dependent seizure suppression, we employed additional Gal80 transgenes that were constitutively expressed in specific subgroups of LK neurons and inhibited Gal4 activity to restrict their expression of the *Indy* RNAi transgene. This intersectional strategy revealed that ABLK neurons in the ventral nerve cord, targeted by *tsh*-Gal80 [57], were dispensable for BSS control (Fig 4A and 4C). In contrast, INDY depletion in a single pair of LHLK neurons, specifically targeted by a glutamatergic *VGlut*-Gal80 transgene, was necessary to induce BSS (Fig 4A and 4C).

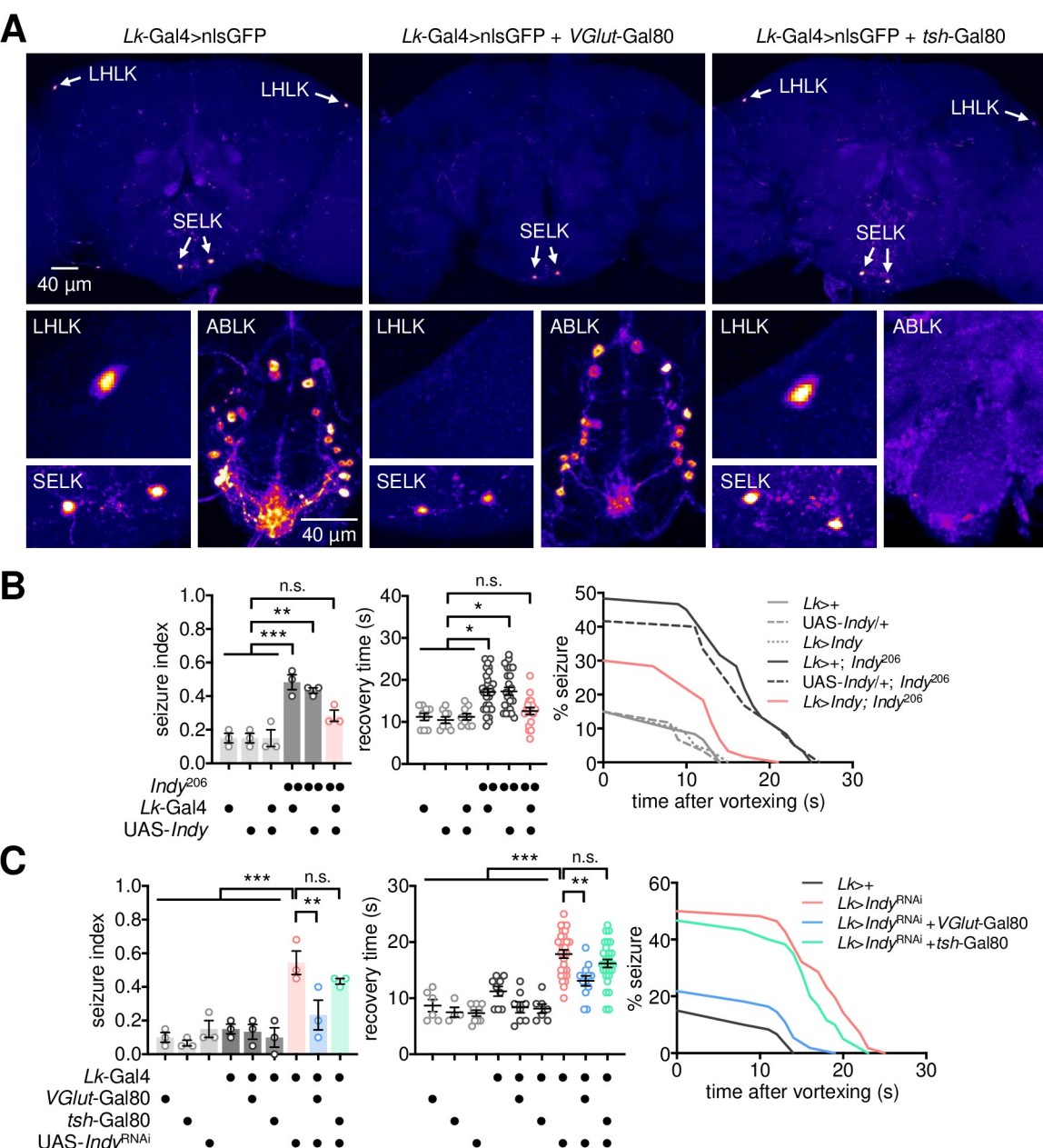

**Fig 4. A pair of LK neurons mediates *Indy*-dependent BSS.** (A) A glutamatergic transgene specifically targets a pair of LHLK neurons among LK neurons. LK neurons in the adult brain (LHLK and SELK) and ventral nerve cord (ABLK) were visualized by transgene expression of nuclear GFP (nlsGFP). The VGlut-Gal80 transgene suppressed GFP expression only in LHLK neurons. Representative confocal images of each genotype were shown. (B) INDY overexpression in LK neurons is sufficient to rescue BSS phenotypes in *Indy* mutants. Quantitative analyses of BSS in individual flies were performed as described in Fig 1. Data represent means ± SEM. n.s., not significant; *$P < 0.05$, **$P < 0.01$, ***$P < 0.001$, as determined by two-way ANOVA with Holm-Sidak's multiple comparisons test (seizure index, n = 60 flies in 3 independent experiments) or by Aligned ranks transformation ANOVA with Wilcoxon rank sum test (recovery time, n = 9–29). (C) INDY depletion in a pair of LHLK neurons is necessary for BSS phenotypes in *Indy* RNAi flies. LK neuron-specific expression of the *Indy* RNAi transgene was inhibited specifically in LHLK neurons by the VGlut-Gal80 transgene. Data represent means ± SEM (seizure index, n = 55–60 flies in 3 independent experiments; recovery time, n = 3–28 flies). n.s., not significant; **$P < 0.01$, ***$P < 0.001$, as determined by one-way ANOVA with Holm-Sidak's multiple comparisons test.

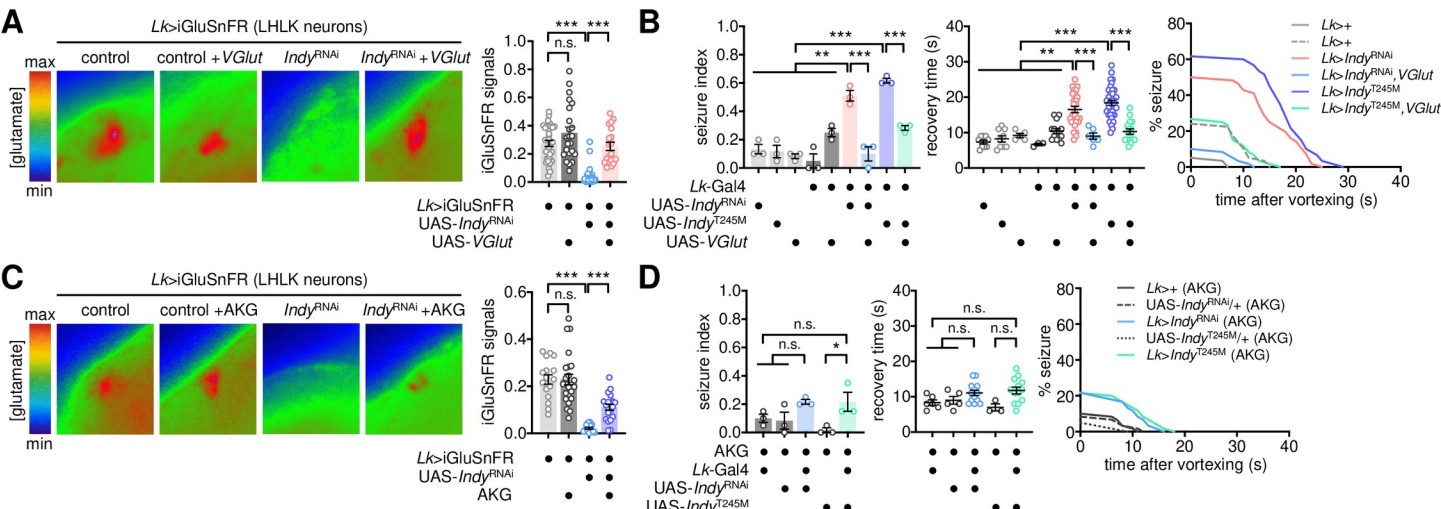

**Fig 5. LK neurons suppress seizures via metabolic control of glutamate transmission.** (A) Overexpression of wild-type VGLUT rescues low glutamate levels in INDY-depleted LHLK neurons. The fluorescent glutamate sensor, iGluSnFR, was co-expressed with *Indy*^RNAi and wild-type VGLUT transgenes in LK neurons. Fluorescence images of LHLK neurons in dissected brains were recorded using photoactivated localization microscopy and analyzed using ZEN software. Relative fluorescence was calculated by normalizing to backgrounds and was averaged for each genotype (n = 19–38). A significant *Indy* x *VGlut* interaction effect on the iGluSnFR signal (*P* < 0.0001) was detected by Aligned ranks transformation ANOVA. Error bars indicate SEM. n.s., not significant; ***P* < 0.001, as determined by Wilcoxon rank sum test. (B) Overexpression of wild-type VGLUT suppresses BSS caused by the loss of *Indy* function in LK neurons. Quantitative analyses of BSS in individual flies were performed as described in Fig 1. Data represent means ± SEM (seizure index, n = 56–60 flies in 3 independent experiments; recovery time, n = 3–37 flies). ***P* < 0.01, ****P* < 0.001, as determined by one-way ANOVA with Holm-Sidak's multiple comparisons test. (C) Oral administration of AKG rescues low glutamate levels in INDY-depleted LHLK neurons. Transgenic flies were fed control or AKG-containing food (20 mM) for 3 d before live-brain imaging. A significant *Indy* x AKG interaction effect on the iGluSnFR signal (*P* = 0.0014) was detected by Aligned ranks transformation ANOVA. Data represent means ± SEM (n = 16–24). n.s., not significant; ****P* < 0.001, as determined by Wilcoxon rank sum test. (D) Oral administration of AKG suppresses BSS caused by the loss of *Indy* function in LK neurons. Data represent means ± SEM (seizure index, n = 60 flies in 3 independent experiments; recovery time, n = 3–13 flies). n.s., not significant; *P* < 0.05, as determined by one-way ANOVA with Holm-Sidak's multiple comparisons test.

A previous study suggested that LK neurons are unlikely glutamatergic [56]. However, we found additional evidence supporting that LHLK neurons indeed express the glutamatergic marker *VGlut*. LK neuron-specific expression of B3 recombinase led to the genomic excision of *VGlut* coding sequence flanked by the B3 recombination target sites from the genome-edited *VGlut* allele (B3RT-*VGlut*-B3RT-LexA) [58], thereby driving the downstream expression of a transgenic LexA driver only in *VGlut*-expressing LK neurons (S9A Fig). Assessment of the transgene expression in the adult fly brain revealed that LHLK neurons, but not SELK neurons, expressed the *VGlut*-derived LexA (S9B Fig), indicating *VGlut* expression only in LHLK neurons. Given these observations, we asked whether *Indy* controls the glutamate transmission from LHLK neurons for seizure suppression. A transgenic fluorescence sensor for detecting synaptic release of glutamate [59] validated that INDY depletion in LHLK neurons lowered the levels of glutamate release (Fig 5A). Transgenic overexpression of wild-type VGLUT not only rescued the glutamate transmission in INDY-depleted LHLK neurons (Fig 5A) but also suppressed their BSS phenotypes (Fig 5B). These observations indicate that the glutamate transmission from INDY-depleted LHLK neurons is likely limiting for seizure suppression.

To validate the *Indy*-relevant link between the TCA cycle and glutamate transmission in seizure-suppressing LHLK neurons, we performed two additional experiments. First, oral administration of α-ketoglutarate partially but significantly restored the glutamate transmission in INDY-depleted LHLK neurons (Fig 5C) and suppressed the BSS phenotypes induced by loss of *Indy* function in LK neurons (Fig 5D). Second, IDH3 depletion in LK neurons was sufficient to induce BSS (S10 Fig), phenocopying the seizure induction by the pan-

glutamatergic IDH3 depletion (S6 Fig). Taken together, these results demonstrate that a pair of glutamatergic LK neurons mediates *Indy*-dependent seizure suppression.

## LK neuron activity gates the behavioral output from seizure initiation

LK signaling has been implicated in neural physiology and behaviors relevant to metabolism in *Drosophila* [55,57,60–64]. It has also been shown that *Indy* mutant flies physiologically mimic calorie-restricted animals and exhibit low fat levels [16], raising the possibility that energy deficiency might underlie *Indy* mutant seizure. Nonetheless, we found some evidence against this idea. First, a lipid-rich diet did not rescue *Indy* mutant seizures while significantly suppressing BSS phenotypes in other seizure mutants [37] (S11A Fig). We note, however, the lipid-rich diet did not rescue low glucose levels in *Indy* mutants (S11B Fig). Second, LK neuron-specific loss of *Indy* function did not affect the baseline levels of triglyceride and glucose (S11C Fig) and the corresponding seizure was insensitive to the lipid-rich supplement (S11D Fig). Finally, hypomorphic mutants of *Lk* and *Lk* receptor (*Lkr*) genes did not exhibit any detectable phenotypes in seizure index under our experimental conditions (S12 Fig). We thus reasoned that LK neuron activity, but not the LK signaling or metabolism per se, might be critical for *Indy*-dependent seizure suppression. Live-brain imaging revealed that INDY depletion reduced the baseline levels of intracellular $Ca^{2+}$ in LHLK neurons (Fig 6A), possibly reflecting their low activity. We asked whether transgenic manipulations of LK neuron activity could either suppress or mimic INDY-depletion phenotypes in BSS.

To this end, optogenetic transgenes were combined with different genetic tools to excite or silence LK neuron activity only during seizure induction for the assessment of their effects on BSS (Fig 6B). Transgenic flies expressing CsChrimson [65] were exposed to red light to transiently excite LK neurons during a mechanical stimulus. This manipulation significantly suppressed BSS phenotypes in *Indy* RNAi flies (Fig 6C), whereas the light transition per se negligibly affected *Indy*-dependent seizure. These observations validate that *Indy* RNAi phenotypes are readily reversible in adult LK neurons within a range of seconds, excluding the possibility that any long-term effects of INDY depletion on neuronal development or metabolic stress contribute to the seizure control. We consistently found that amber light-dependent silencing of LHLK neurons by the eNpHR transgene [66] was sufficient to induce BSS, even in a wild-type background (Fig 6D). These results demonstrate that LK neuron activity inversely correlates with seizure susceptibility. Nonetheless, the optogenetic inhibition of LK neuron activity alone did not trigger seizure-like behaviors in the absence of a mechanical stimulus (S13 Fig). We reason that this neural locus may not initiate seizure-like behaviors but inhibit the behavioral output from an as-yet-unmapped seizure-initiating locus in the adult *Drosophila* brain.

## NMDA receptors in dorsal fan-shaped body neurons act downstream of LHLK neurons to suppress BSS

Although the implication of LK-LKR signaling in *Indy*-dependent seizure was not evident, we reasoned that LKR-expressing neurons could still act downstream of LHLK neurons via the seizure-suppressing glutamate transmission. It has been shown that LKR neurons are present in distinct parts of the adult fly brain, including the pars intercerebralis, ellipsoid body, and fan-shaped body (FSB) [55,57,60]. Also, our transgenic expression of the GFP-fused synaptotagmin 1 revealed LHLK projections in the LH and FSB (Fig 7A). We thus employed a dorsal FSB (dFSB)-specific Gal4 driver (i.e., R23E10-Gal4) to examine whether glutamate receptors expressed in dFSB neurons contribute to the susceptibility of

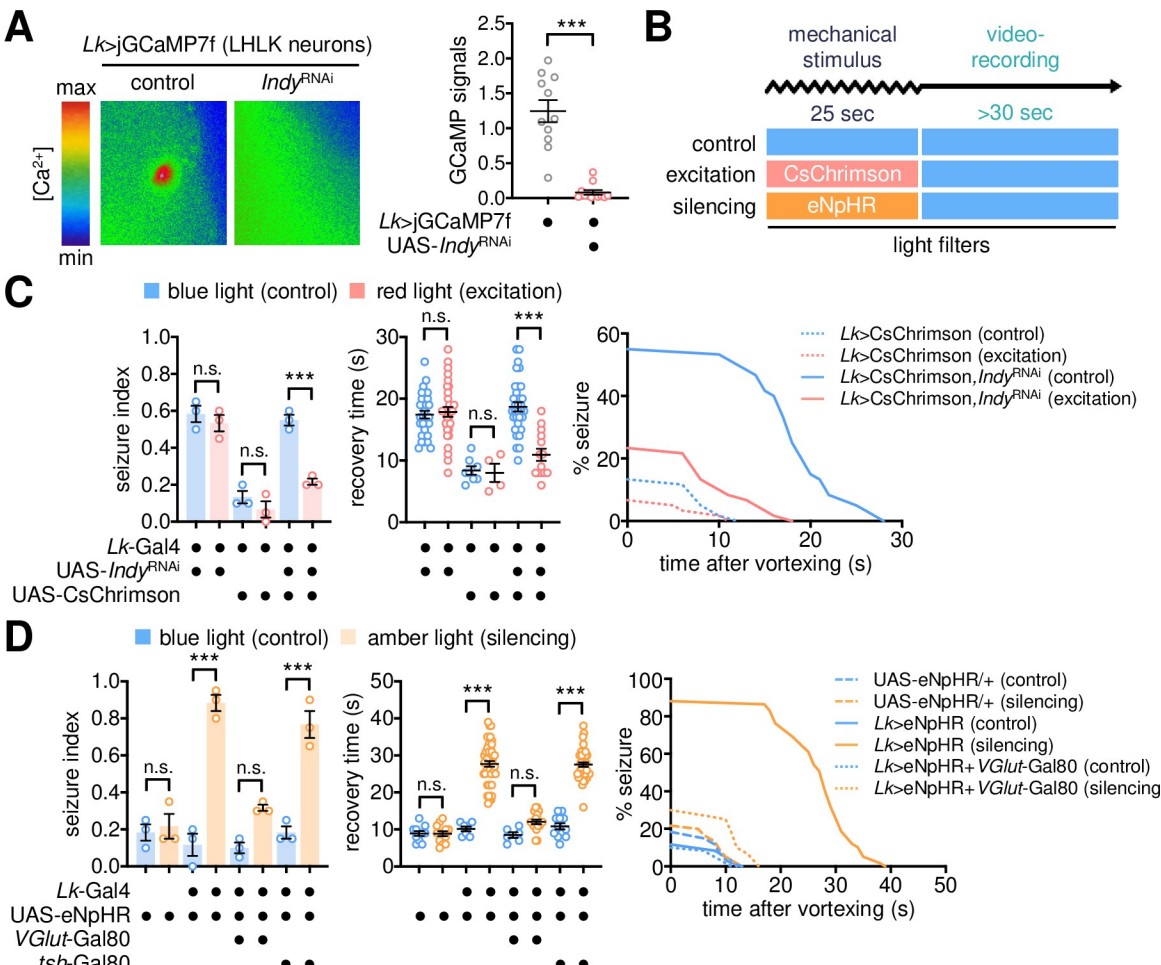

**Fig 6. LK neuron activity gates the behavioral output from seizure initiation.** (A) INDY depletion reduces intracellular $Ca^{2+}$ levels in LHLK neurons. The codon-optimized calcium sensor, jGCaMP7f, was co-expressed with $Indy^{RNAi}$ in LK neurons. The fluorescence signal in LHLK neurons was analyzed as described in Fig 5A. Data represent means ± SEM (n = 11–12). ***$P < 0.001$, as determined by Mann-Whitney test. (B) Experimental design for optogenetic manipulations of neural activity during the assessment of BSS. (C) Optogenetic excitation of LK neurons substantially suppresses BSS in INDY-depleted flies. Transgenic flies were crossed and kept in constant darkness. The mechanical stimulus was given under blue light (no excitation) or red light (excitation by CsChrimson), and BSS phenotypes were then assessed under blue light. Quantitative analyses of BSS in individual flies were performed as described in Fig 1. Data represent means ± SEM (seizure index, n = 56–60 flies in 3 independent experiments; recovery time, n = 4–33 flies). n.s., not significant; ***$P < 0.001$, as determined by two-way ANOVA with Holm-Sidak's multiple comparisons test. (D) Optogenetic silencing of glutamatergic LK neurons induces BSS in a wild-type background. The mechanical stimulus was given under blue light (no silencing) or amber light (silencing by eNpHR). Data represent means ± SEM (seizure index, n = 59–60 flies in 3 independent experiments; recovery time, n = 6–52 flies). n.s., not significant; ***$P < 0.001$, as determined by two-way ANOVA with Holm-Sidak's multiple comparisons test.

BSS. We indeed found that dFSB-specific depletion of NMDAR2 was sufficient to cause BSS phenotypes, comparable to the high seizure index and long recovery time observed in *Nmdar2* mutants (Fig 7B). Neither electrical silencing of dFSB neurons by the inwardly rectifying Kir2.1 channel [67] nor blocking their synaptic transmission by tetanus toxin light chain (TNT) [68] significantly affected seizure index; however, both the transgenic manipulations lengthened the recovery time in BSS-positive animals (Fig 7C). These data together map the metabolic control of seizure onset to the glutamate transmission between LHLK and dFSB neurons and suggest that the subsequent output of dFSB neurons may govern the duration of seizure-like behaviors.

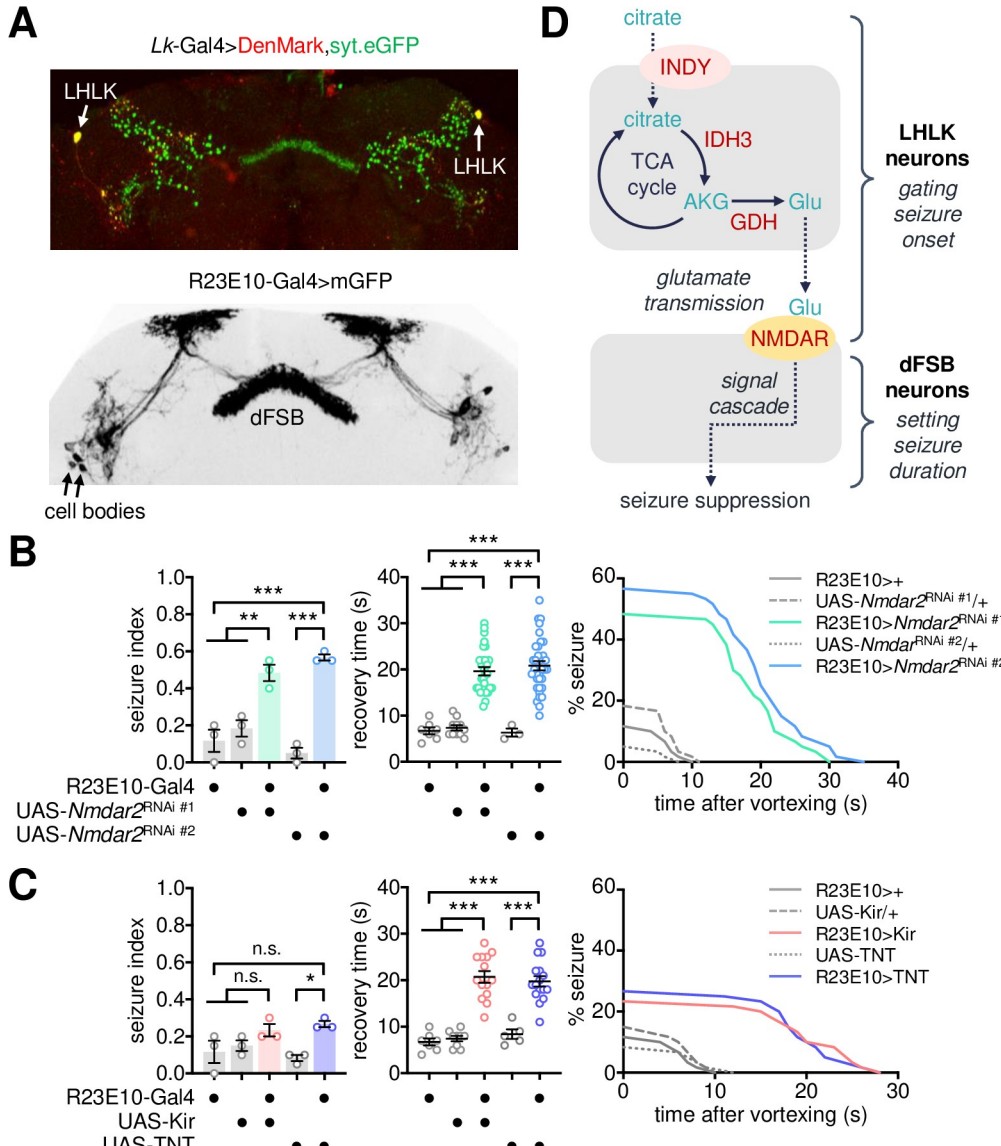

**Fig 7. NMDA receptors in dorsal fan-shaped body neurons relay seizure-suppressing glutamate transmission.** (A) Representative confocal images of dendrites (DenMark, red) and axonal projections (syt.eGFP, green) from LHLK neurons (top) and mGFP expression from dFSB-specific R23E10-Gal4 driver (bottom) in the adult fly brain. (B) Depletion of NMDA receptors in dFSB neurons induces BSS. Quantitative analyses of BSS in individual flies were performed as described in Fig 1. Data represent means ± SEM. $^{**}P < 0.01$, $^{***}P < 0.001$, as determined by one-way ANOVA with Holm-Sidak's multiple comparisons test (seizure index, n = 60 in 3 independent experiments) or by Welch's ANOVA with Dunnett's multiple comparisons test (recovery time, n = 3–34). (C) Genetic silencing of dFSB neurons lengthens recovery time after BSS. Data represent means ± SEM. n.s., not significant; $^{*}P < 0.05$, $^{***}P < 0.001$, as determined by one-way ANOVA with Holm-Sidak's multiple comparisons test (seizure index, n = 60 in 3 independent experiments; recovery time, n = 5–16 for TNT) or by Welch's ANOVA with Dunnett's multiple comparisons test (recovery time, n = 7–14 for Kir). (D) A working model for the neurometabolic pathway of *Indy*-dependent seizure suppression.

## Discussion

Human genetic studies have previously identified several loss-of-function alleles of *ROGDI* or *SLC13A5* among KTS-associated loci. In the current study, we behaviorally assessed the susceptibility to a specific form of seizure-like activity in *Drosophila* mutants of the KTS-

associated genes. We defined genetic, biochemical, and neural pathways of *SLC13A5*/*Indy*-relevant seizure in the context of the citric acid cycle, glutamate metabolism, and glutamatergic transmission (Fig 7D). We further mapped the reversible and scalable functions of the seizure suppressor gene to a surprisingly small group of glutamatergic neurons (i.e., a pair of LHLK neurons) in the adult fly brain. In a broad sense, these findings demonstrate a specific micro-neural circuit that gates both initiation and duration of seizure-like behaviors.

Given the citrate transporter activity of SLC13A5/INDY, it has been suggested that neuronal energy failure may cause *SLC13A5*-associated epilepsy [7,8]. Citrate acts as a precursor of fatty acid biosynthesis while also serving as a key intermediate in the TCA cycle. Accordingly, an *Indy* mutation mimics calorie-restricted conditions and thereby causes metabolic and longevity phenotypes in *Drosophila* and mouse models, including alterations in lipid metabolism (i.e., low-fat storage), insulin signaling, and mitochondrial biogenesis [16,21]. Perturbations of the TCA cycle and altered levels of extracellular TCA metabolites are consistently observed in *SLC13A5*-associated KTS patients [69]. In fact, genetic losses of other metabolic enzymes involved in the TCA cycle (e.g., IDH3, malate dehydrogenase, fumarase hydratase) have been implicated in several neurological disorders, including early-onset seizure and developmental delay [70–74]. Nonetheless, there are conflicting reports on the effects of a ketogenic diet on KTS-associated epilepsy [8].

Genetic or pharmacological manipulations of metabolic enzymes and relevant genes likely impact on general cell physiology. The long-term metabolic stress may thus lead to pleiotropic effects and poor phenotypic outcomes, possibly explaining the seizure phenotypes observed in our genetic models. Nonetheless, several lines of our evidence support a more specific mechanism for *Indy*-dependent seizure suppression. First, *Indy* mutants did not display any gross motor defects, while their seizure phenotypes were mapped very specifically to a small group of the glutamatergic LK neurons among other broadly defined groups of neurons (e.g., GABAergic or cholinergic neurons). Second, *Indy* displayed specific genetic interactions with *Idh3*, *VGlut*, and select glutamate receptors on BSS in trans-heterozygous conditions, whereas BSS were not detected in individual heterozygous mutants. Third, genetic enhancement of the glutamate transmission by VGLUT overexpression in LK neurons was sufficient to suppress BSS in INDY-depleted flies. Finally, the INDY-depletion phenotypes were readily reversible in the adult LK neurons (i.e., AKG administration, conditional RNAi) and the optogenetic excitation of INDY-depleted LK neurons only during a mechanical stimulus was sufficient to suppress BSS although the possible off-target effects of the *Indy* RNAi transgene were not completely excluded. These observations together support our model that impairment of the metabolic flux between the TCA cycle and glutamate biosynthesis—specifically, the conversion of α-ketoglutarate to glutamate—down-scales the seizure-suppressing transmission from a specific subset of glutamatergic neurons, resulting in BSS phenotypes.

Previous studies in animal models have actually implicated low levels of TCA metabolites and glutamate in epilepsy [73,75–79]. Moreover, co-injection of α-ketoglutarate suppresses chemically induced epilepsy in mice [80,81], consistent with our results. Intriguingly, the recent observation that α-ketoglutarate promotes the interaction between synaptotagmin 1 and phospholipids demonstrates an unexpected role of α-ketoglutarate in synaptic vesicle fusion [51]. Thus, we reasoned that an excess of α-ketoglutarate might deplete synaptic vesicle pools and thereby interfere with their transmission. This explains why oral administration of α-ketoglutarate to wild-type flies induces rather than suppresses BSS under our experimental conditions.

Although *Drosophila* genetic studies have led to the isolation and characterization of a number of seizure-related mutants, much less is known about the neural loci responsible for inducing seizure and sustaining seizure-like activity [82]. For instance, the seizure-suppressing

function of the potassium-chloride symporter *kazachoc* and seizure-inducing neural locus in a BSS mutant of the voltage-gated sodium channel *paralytic* have been mapped to the mushroom body in the adult brain [83,84]. Seizure suppression by the product of the transmembrane domain gene, *julius seizure*, was more broadly mapped to cholinergic or GABAergic neurons but not to glutamatergic neurons [85]. The emergence of a seizure has been thought to involve three distinct components of neural circuits that mediate seizure initiation, seizure buildup, and seizure spread, respectively [86]. Modulatory neurons likely regulate the excitability of these neural components and thus shape seizure intensity or duration. Based on this model, we propose that glutamatergic LK neurons mediate a seizure-inhibitory pathway such that their conditional silencing de-represses both seizure onset and duration. Accordingly, down-regulation of the glutamatergic transmission from LK neurons induces BSS, as opposed to the general involvement of excitatory glutamate in seizure induction [44–46].

Our data suggest that LK signaling is not directly involved in controlling seizure susceptibility, which makes sense considering the temporal scale of neuropeptide signaling in general. Nonetheless, we found that the metabolic flux of the TCA cycle in LK neurons and their glutamate transmission onto dFSB neurons might be closely linked to the control of BSS. Evidence for the involvement of LK neurons in feeding, metabolism, and associated physiology (e.g., circadian behaviors, sleep, memory) is abundant [55,57,60–64]. The activity of LHLK neurons is also sensitive to the metabolic state and the time of day [57,60,63]. Thus, we hypothesize that LK neurons act as a neural sensor that integrates internal cues (e.g., circadian clocks, sleep state, and metabolic state) while engaging different postsynaptic partners and divergent pathways for relevant physiological outputs through distinct signaling molecules. In particular, dFSB neurons are a key sleep-promoting locus in the adult fly brain, analogous to the sleep-promoting ventrolateral preoptic nucleus (VLPO) in mammals [87,88]. Given the implication of VLPO in consciousness loss during epileptogenesis [89], the LK-dFSB pathway may reveal the conserved neural principles underlying intimate interactions among metabolism, epilepsy, and sleep [90–92].

*Drosophila* models of human genetic disorders have proven to be valuable tools for elucidating underlying pathogenic mechanisms. Our findings enrich this body of knowledge by providing a genetic, biochemical, and neural map of the seizure suppressor pathway related to KTS. The complexity of the mammalian brain and species-specific organization of excitatory neurotransmitters in the nervous systems may limit the direct relevance of our working model to KTS pathogenesis at neural-circuit levels. Nonetheless, given the strong conservation of *SLC13A5/Indy* homologs and their physiological function among different species, we propose that similar cellular mechanisms may underlie early-onset seizures in KTS patients and explain their relative resistance to generic antiepileptic drugs.

## Materials and methods

### *Drosophila* stocks

All flies were maintained in standard cornmeal–yeast–agar medium at room temperature. $w^{1118}$ (BL5905; a wild-type control), $Indy^{206}$ (BL27901), $Indy^{302}$ (BL27902), $Idh3g^{G9423}$ (BL30194; CG5028), $Lk^{C275}$ (BL16324), $Lkr^{MI06336}$ (BL41520), $Lkr^{MI08640}$ (BL51094), $Nmdar1^{EP331}$ (BL17331), $Nmdar2^{attP}$ (BL84548), $GluRIA^{attP}$ (BL84506), $GluRIB^{attP}$ (BL84507), $mGluR^{MI02169}$(BL32830), $GluClα^{glc1}$(BL6353), VGlut$^{OK371}$-Gal4 (BL26160), R23E10-Gal4 (BL49032), VGlut-Gal80 (BL58448), tub-Gal80$^{ts}$ (BL7018), 20XUAS-eNpHR3.0.YFP (BL36350), 20XUAS-IVS-CsChrimson.mVenus (BL55135), 20XUAS-iGluSnFR.A184A (BL59609), UAS-Denmark, UAS-syt.eGFP (BL33065), UAS-$Nmdar2^{RNAi}$ (#1, BL26019), UAS-$Nmdar2^{RNAi}$ (#2, BL40846), 20XUAS-IVS-jGCaMP7f (BL80906), Tdc2[B3RT]-LexA;

UAS-B3R, LexAop2-His2B-mCherry, UAS-His2A-GFP (BL91248), and VGlut[B3RT]-LexA (BL91249) were obtained from Bloomington Drosophila Stock Center. UAS-*Indy*[RNAi] #1 (3979R-1), UAS-*Indy*[RNAi] #2 (3979R-3), UAS-*Idh3a*[RNAi] (12233R-3), UAS-*Idh3b*[RNAi] (6439R-2), and UAS-*Idh3g*[RNAi] (5028R-2) were obtained from National Institute of Genetics. UAS-*Indy*[RNAi] #3 (v9981), UAS-*Indy*[RNAi] #4 (v9982), UAS-*rogdi*[RNAi] (v107310) was obtained from Vienna Drosophila Resource Center. *para*[bss], *eas*[2], and *sda*[iso7.8] were gifts from Dr. Kitamoto [36] and Dr. Kuebler [37]. *VGlut*[1], *VGlut*[2], and UAS-*VGlut* were gifts from Dr. DiAntonio [40]. *rogdi*[del], *rogdi*[P1], UAS-*rogdi*-3xFLAG, UAS-Kir2.1, UAS-TNT, Lk-Gal4, and tsh-Gal80 have been described previously [32,56,67,68,93]. Indy-PB cDNA was PCR-amplified from a cDNA clone (RH67364, Drosophila Genomics Resource Center) and inserted into pUAST-attB-V5 [94]. Subsequently, a T[ACT]>M[ATG] mutation was introduced to the wild-type cDNA to mimic the KTS-associated 680C>T allele. UAS-*Indy* and UAS-*Indy*[T245M] transgenic lines were generated by site-specific transformation on the attP40 landing site (BestGene Inc.)

## Bang-sensitive seizure analysis

To avoid any genetic-background effects on bang-sensitive seizure, *Indy* mutant stocks were outcrossed six times to the wild-type control (*w*[1118]) for isogenization. All the behavioral tests for heterozygous, homozygous, and trans-heterozygous mutants were conducted with the iso-genized *Indy* lines and *w*[1118] control. For transgenic lines, seizure behaviors in trans-heterozygous animals were compared to those in all appropriate heterozygous controls. Where applicable, additional transgenic combinations or independent transgenes were tested for validation. Unless otherwise indicated, BSS was assessed in male flies. Three to seven-day-old flies were fed cornmeal–yeast–agar medium in LD cycles at 25˚C for 3 d. They were $CO_2$-anesthe-tized and harvested into fresh vials containing the same food >3 h before a group of flies (n = 5 for non-optogenetic experiments; n = 5–10 for optogenetic experiments) was trans-ferred to an empty vial using an aspirator. Each vial was vortexed at a maximum speed for 25 s using Vortex-Genie 2 (Scientific Industries, Inc.) and then video-recorded for >30 s using a cellular phone (LG G5 Pro or Samsung Galaxy Note 8). BSS-positive flies were defined if they displayed initial paralysis upon mechanical stimulus, followed by uncoordinated movements such as wing-flapping, abdominal contractions, or leg-twitching before their recovery of a nor-mal posture. BSS was quantitatively assessed by three parameters. A seizure index was calcu-lated as the ratio of the number of BSS-positive flies to the total number of flies tested in each experiment (n = ~20 flies per genotype or condition) and averaged from three independent experiments. Recovery time was calculated individually for BSS-positive flies as the latency to normal posture after vortexing and was averaged for each genotype or condition. Percent sei-zure was calculated as the percentage of BSS-positive flies at each second after vortexing (n = ~60 flies per genotype or condition from all the three experiments). To determine the length of a refractory period in *Indy* mutants, the first mechanical stimulus was given to each fly and then the second mechanical stimulus was given only to BSS-positive animals at the indicated time after recovery from their first BSS. The seizure index was calculated in each experiment (n = 10 flies per time point) and averaged from three independent experiments.

## Locomotor behavior analysis

Locomotor activities in individual male flies were measured in 1-min bins using the *Drosophila* Activity Monitor system (TriKinetics). Daily locomotor activity (total activity count per 12-hour light: 12-hour dark cycle) and waking activity (averaged activity count per minute awake) were analyzed as described previously [95,96]. For the climbing activity measurement, a group of 10 male flies was kept in the climbing chamber. Flies were allowed to climb for 5

seconds after gentle tapping-down. Climbing distance in each fly was measured as the highest position during the recording.

## Oral administration of chemicals

Each chemical was dissolved in 5% sucrose and 2% agar food (behavior food) at the indicated concentration and fed to flies for 3 d before the BSS assessment. The chemicals tested included epigallocatechin gallate (EGCG, Sigma), diethylstilbestrol (DES, Sigma), α-ketoglutarate (Sigma), and KetoCal 4: 1 (fat: carbohydrate plus protein, Nutricia).

## Optogenetics

Transgenic flies were grown in light-tight vials and kept in constant dark. They were loaded on to the behavior food containing 0.1 mM all-trans retinal (Sigma) and entrained at 25°C for 3 d. A light source with the indicated wavelength was turned on during the mechanical stimulation. A white LED was covered with a 585-nm or 470-nm filter to generate amber or blue light, respectively, while the 630-nm LED provided red light.

## Immunofluorescence imaging

Immunostaining of whole-mount adult brains was performed as described previously [97]. Dissected brains were fixed with 3.7% formaldehyde in PBS, blocked with PBS containing 0.3% Triton X-100 and 5% normal goat serum, and incubated with anti-GFP antibody (Invitrogen, A-11122; diluted at 1:1000) or anti-mCherry antibody (Developmental Studies Hybridoma Bank, DSHB-mCherry-3A11; diluted at 1:20) at 4°C overnight. After washing in PBS containing Triton X-100, immunostained samples were further incubated with species-specific Alexa Fluor secondary antibodies (Jackson ImmunoResearch) at 4°C overnight, washed with PBS containing Triton X-100, and then mounted using VECTASHIELD mounting medium (Vector Laboratories). Confocal images were obtained using an FV1000 microscope (Olympus) and analyzed using ImageJ software.

## Live-brain imaging

Transgenic flies were loaded on to the behavior food containing 0 mM (control) or 20 mM α-ketoglutarate (Sigma), entrained at 25°C for 3 d, and then anesthetized in ice. Whole brains were dissected out in hemolymph-like HL3 solution (5 mM HEPES pH 7.2, 70 mM NaCl, 5 mM KCl, 1.5 mM CaCl$_2$, 20 mM MgCl$_2$, 10 mM NaHCO$_3$, 5 mM trehalose, 115 mM sucrose) and then transferred on cover glass in a magnetic imaging chamber (Chamlide CMB, Live Cell Instrument) filled with HL3 buffer. Fluorescence images from live brains were recorded at room temperature using photoactivated localization microscopy (ELYRA P.1, Carl Zeiss) with a C-Apochromat 40x/1.20 W Korr M27 at a pixel resolution of 512 x 512. Fluorescence signals were quantified by background subtraction and analyzed using ZEN software (Carl Zeiss).

## Quantitative transcript analysis

Total RNA was purified from thirty fly heads using TRIzol reagent (Thermo Fisher Scientific). After DNase I digestion, cDNA was synthesized using M-MLV Reverse Transcriptase (Promega) and random hexamers. Quantitative real-time PCR was performed using Prime Q-Mastermix (GeNet Bio) and LightCycler 480 Instrument II (Roche). The qPCR primers used in this study were as follows: 5'-TTC ATC GCT TCA CGT CAC TC-3' (forward) and 5'-TGC TGA CTT GGT GGA TTT TG-3' (reverse) for *rogdi*; 5'-ATC TCC CAC AGG ACG TCA AC-

3' (forward) and 5'-GCG ACG AAG AGA AGG ATC AC-3' (reverse) for *pabp* (internal control). *Indy* primers have been described previously [16].

## Amino acid profiling

Male flies were loaded on to behavior food containing either 0 mM (control) or 20 mM α-ketoglutarate and then entrained in LD cycles at 25˚C for 3 d before harvest. Whole-body extracts were prepared from 50 flies per condition, and the levels of free amino acids were quantitatively measured using ion-exchange chromatography as described previously [95,96].

## Metabolite measurements

Whole-body extracts were prepared by homogenizing 8 male flies per sample in 200 μl of PBS containing 0.05% Tween-20. Triglyceride and glucose measurements were performed using Triglyceride Reagent (Sigma-Aldrich, T2449), Free Glycerol Reagent (Sigma-Aldrich, F6428), and Glucose Assay Reagent (Sigma-Aldrich, G3293), respectively, according to the manufacturers' instructions.

## Statistical analysis

Statistical analyses were performed using GraphPad Prism, R (version 3.5.3), or Microsoft Excel. Shapiro-Wilk and Brown-Forsythe tests were conducted to check normality ($P < 0.05$) and equality of variances ($P < 0.05$), respectively. For comparisons among multiple samples: 1) Parametric datasets with equal variance were further analyzed by ordinary ANOVA. 2) Parametric datasets with unequal variance were further analyzed by Welch's ANOVA (one-way) or Aligned ranks transformation ANOVA (two-way). 3) Nonparametric datasets with equal variance were analyzed by Kruskal-Wallis ANOVA (one-way) or Aligned ranks transformation ANOVA (two-way). 4) Nonparametric datasets with unequal variance were analyzed by Aligned ranks transformation ANOVA. Post hoc multiple comparisons were performed by Holm-Sidak's (ordinary ANOVA), Dunnett's T3 (Welch's ANOVA), Wilcoxon rank sum (Aligned ranks transformation ANOVA), or Dunn's test (Kruskal-Wallis ANOVA). For comparisons between two samples, nonparametric datasets were analyzed by Mann-Whitney U test. Datasets including any sample size less than 7 were analyzed by Student's t-test or ordinary ANOVA. Sample sizes and *P* values obtained from individual statistical analyses were indicated in the figure legends.

## Supporting information

**S1 Fig. *Indy* mutant flies display BSS comparably to other seizure mutants.** (A) The seizure phenotypes in *Indy* mutants were comparable to those observed in *easily shocked* (*eas*[2]) and *slamdance* (*sda*[iso7.8]) but weaker than *bang senseless* (*para*[bss]) mutants. Quantitative analyses of BSS in individual flies were performed as described in Fig 1. Data represent means ± SEM (seizure index, n = 60 flies in 3 independent experiments; recovery time, n = 2–59 flies). n.s., not significant; *$P < 0.05$, ***$P < 0.001$, as determined by one-way ANOVA with Holm-Sidak's multiple comparisons test. (B) *Indy* mutant seizure displays a refractory period after seizure recovery. The first mechanical stimulus was given to each fly and then the second mechanical stimulus was given only to BSS-positive animals at the indicated time after recovery from their first BSS. The seizure index was calculated in each experiment (n = 10 flies per time point) and averaged from three independent experiments. Error bars indicate SEM.
(TIF)

**S2 Fig. Loss of *Indy* function in specific neurons induces BSS.** (A) Pan-neuronal overexpression of *Indy*^RNAi transgene reduces endogenous *Indy* expression. Total RNA was purified from fly heads. Quantitative analyses of *Indy* and poly(A)-binding protein (normalizing control) mRNAs were performed using real-time PCR with gene-specific primer sets. The relative levels of *Indy* mRNA in each genetic background were calculated by normalizing to *Elav>Dcr2* control (set as 1). Data represent means ± SEM (n = 3). ***$P < 0.001$ as determined by one-way ANOVA with Holm-Sidak's multiple comparisons test. (B) A genetic screen identifies *VGlut*- and *Lk*-expressing neurons as neural loci important for *Indy*-dependent control of the seizure susceptibility. Quantitative analyses of BSS in individual flies were performed as described in Fig 1. Data represent means ± SEM (n = 60 flies in 3 independent experiments). **$P < 0.01$, ***$P < 0.001$, as determined by one-way ANOVA with Holm-Sidak's multiple comparisons test. (C) The BSS induction by glutamatergic INDY depletion is consistently observed by independent *Indy* RNAi transgenes. Data represent means ± SEM (seizure index, n = 60 flies in 3 independent experiments; recovery time, n = 1–38 flies). *$P < 0.05$, **$P < 0.01$, ***$P < 0.001$, as determined by one-way ANOVA with Holm-Sidak's multiple comparisons test. (TIF)

**S3 Fig. Multiple sequence alignment of *SLC13A5* homologs reveals the evolutionary conservation of an amino acid residue associated with KTS patients.** *Drosophila Indy*^T245M mutant mimics the gene product of the KTS-associated 680C>T allele. TM, a transmembrane region. (TIF)

**S4 Fig. Loss of *rogdi* function in glutamatergic neurons induces BSS.** (A) *rogdi* mutants homozygous or trans-heterozygous for loss-of-function alleles display BSS. Quantitative analyses of BSS in individual flies were performed as described in Fig 1. Data represent means ± SEM. **$P < 0.01$, ***$P < 0.001$ as determined by one-way ANOVA with Holm-Sidak's multiple comparisons test (seizure index, n = 60 flies in 3 independent experiments) or by Aligned ranks transformation ANOVA with Wilcoxon rank sum test (recovery time, n = 7–29 flies). (B) Pan-neuronal overexpression of *rogdi*^RNAi transgene reduces endogenous *rogdi* expression in fly heads. Quantitative transcript analyses were performed as described in S2A Fig. Data represent means ± SEM (n = 3). ***$P < 0.001$ as determined by one-way ANOVA with Holm-Sidak's multiple comparisons test. (C) ROGDI depletion in glutamatergic neurons is sufficient to induce BSS. Quantitative analyses of BSS in individual flies were performed as described in Fig 1. Data represent means ± SEM (seizure index, n = 60 flies in 3 independent experiments; recovery time, n = 5–27 flies). ***$P < 0.001$, as determined by one-way ANOVA with Holm-Sidak's multiple comparisons test. (D) Transgenic overexpression of wild-type ROGDI in glutamatergic neurons rescues BSS in *rogdi* mutants. Data represent means ± SEM (seizure index, n = 49–60 flies in 3 independent experiments; recovery time, n = 6–26 flies). n.s., not significant; **$P < 0.01$, ***$P < 0.001$, as determined by two-way ANOVA with Holm-Sidak's multiple comparisons test. (TIF)

**S5 Fig. Oral administration of the GDH inhibitor diethylstilbestrol (DES) exaggerates BSS phenotypes in *Indy* mutants.** Flies were fed control or DES-containing food (10 or 20 μg/mg) for 3 d before the BSS assessment. Quantitative analyses of BSS in individual flies were performed as described in Fig 1. Two-way ANOVA detected a significant *Indy* x DES interaction effect on recovery time ($P = 0.0421$). Data represent means ± SEM (seizure index, n = 60 flies in 3 independent experiments; recovery time, n = 3–49 flies). n.s., not significant; **$P < 0.01$,

***$P < 0.001$, as determined by Holm-Sidak's multiple comparisons test.
(TIF)

**S6 Fig. Depletion of individual IDH3 subunit proteins in glutamatergic neurons induces BSS in a wild-type background.** Quantitative analyses of BSS in individual flies were performed as described in Fig 1. Data represent means ± SEM. **$P < 0.01$, ***$P < 0.001$ as determined by one-way ANOVA with Holm-Sidak's multiple comparisons test (seizure index, n = 60 flies in 3 independent experiments), by Aligned ranks transformation ANOVA with Wilcoxon rank sum test (recovery time, n = 8–43 flies for *Idh3a*<sup>RNAi</sup> or *Idh3b*<sup>RNAi</sup>), or by Welch's ANOVA with Dunnett's T3 multiple comparisons test (recovery time, n = 8–35 flies for *Idh3g*<sup>RNAi</sup>).
(TIF)

**S7 Fig. *Indy* mutant flies do not show general motor defects.** (A) $w^{1118}$ control and *Indy* mutant flies show similar waking activities under 12-h light: 12-h dark (LD) cycles. Individual male flies were transferred to $65 \times 5$ mm glass tubes containing 5% sucrose and 2% agar food and entrained in LD cycles. Locomotor activities were indirectly measured by infrared beam crosses per minute using the Drosophila Activity Monitor system. Daily locomotor activity (activity/day) and waking activity (activity/min awake) were calculated in each fly on the fourth LD cycles and averaged (n = 25 and 29 flies for $w^{1118}$ control and *Indy*$^{206}$ mutants, respectively). Error bars indicate SEM. n.s., not significant; *$P < 0.05$ as determined by Mann-Whitney U test. (B) $w^{1118}$ control and *Indy* mutant flies display comparable climbing activities. A group of 10 male flies was kept in the climbing chamber and then allowed to climb for 5 seconds after gentle tapping-down. Climbing distance in each fly was measured as the highest position during the recording and averaged (n = 30). Error bars indicate SEM. n.s., not significant as determined by Student's t-test.
(TIF)

**S8 Fig. Adult LK neurons mediate *Indy*-dependent BSS.** Adult-specific INDY depletion in LK neurons is sufficient to induce BSS. Transgenic flies were crossed and kept at 21˚C to block the expression of *Indy*<sup>RNAi</sup> transgene using tub-Gal80<sup>ts</sup>. Adult flies were then incubated at 21˚C (no depletion) or 29˚C (RNAi-mediated depletion) for >24 hours prior to the assessment of BSS at the same temperature. Quantitative analyses of BSS in individual flies were performed as described in Fig 1. Data represent means ± SEM (seizure index, n = 60 flies in 3 independent experiments; recovery time, n = 5–28). n.s., not significant; **$P < 0.01$, ***$P < 0.001$, as determined by one-way ANOVA with Holm-Sidak's multiple comparisons test.
(TIF)

**S9 Fig. LHLK neurons express the glutamatergic marker gene *VGlut*.** (A) A transgenic strategy for visualizing *VGlut*-expressing LK neurons by the fluorescent reporter proteins. The CRISPR-edited *VGlut* locus includes two B3 recombination target (B3RT) sites upstream of the LexA-coding sequence. LK neuron-specific overexpression of the B3 recombinase leads to the genomic excision, thereby allowing LexA expression only in *VGlut*-expressing LK neurons. LexA expression could be indirectly visualized by the transgenic mCherry reporter. (B) LHLK neurons, but not SELK neurons, express B3RT-LexA transgene from the *VGlut*-deleted locus. Representative confocal images of LHLK and SELK neurons in the adult fly brain were shown along with the full genotype.
(TIF)

**S10 Fig. IDH3 depletion in LK neurons induces BSS.** Quantitative analyses of BSS in individual flies were performed as described in Fig 1. Data represent means ± SEM (seizure index, n = 44–60 in 3 independent experiments; recovery time, n = 8–27). $^*P < 0.05$, $^{**}P < 0.01$, $^{***}P < 0.001$, as determined by one-way ANOVA with Holm-Sidak's multiple comparisons test (seizure index), by Aligned ranks transformation ANOVA with Wilcoxon rank sum test (recovery time, *Idh3a* and *Idh3b* RNAi), or by Welch's ANOVA with Dunnett's multiple comparisons test (recovery time, *Idh3g* RNAi).
(TIF)

**S11 Fig. *Indy*-relevant seizure unlikely involves metabolic effects.** (A) A lipid-rich diet rescues BSS in *eas*² mutants but not in *Indy*²⁰⁶ mutants. Flies were fed control or 5% KetoCal food for 3 d before the BSS assessment. Quantitative analyses of BSS in individual flies were performed as described in Fig 1. Two-way ANOVA detected a significant *eas* x KetoCal interaction effect on seizure index ($P = 0.0006$). Data represent means ± SEM (seizure index, n = 60 flies in 3 independent experiments; recovery time, n = 4–34). n.s., not significant; $^*P < 0.05$, $^{**}P < 0.01$, $^{***}P < 0.001$, as determined by Holm-Sidak's multiple comparisons test. (B) The lipid-rich diet does not rescue low glucose levels in *Indy* mutants. Flies were fed control or 5% KetoCal (4: 1 = fat: carbohydrate plus protein) food for 3 d before harvesting. Triglyceride and glucose levels in whole-body extracts were quantified using standard curves. Two-way ANOVA detected significant *Indy* x KetoCal interaction effects on triglyceride ($P = 0.0019$) and glucose levels ($P = 0.0228$). Data represent means ± SEM (n = 5–8). n.s., not significant; $^*P < 0.05$, $^{***}P < 0.001$, as determined by Holm-Sidak's multiple comparisons test. (C) Loss of *Indy* function in LK neurons does not cause a change in metabolite levels. Data represent means ± SEM (n = 4–5). n.s., not significant as determined by two-way ANOVA with Holm-Sidak's multiple comparisons test. (D) The lipid-rich diet does not rescue BSS induced by LK neuron-specific loss of *Indy* function. Two-way ANOVA detected no significant *Indy* x KetoCal interaction effect on seizure index ($P = 0.7802$ for *Indy*^RNAi^; $P = 0.1725$ for *Indy*^T245M^) and recovery time ($P = 0.9534$ for *Indy*^RNAi^; $P = 0.7273$ for *Indy*^T245M^). n.s., not significant; $^*P < 0.05$, $^{**}P < 0.01$, $^{***}P < 0.001$, as determined by Holm-Sidak's multiple comparisons test.
(TIF)

**S12 Fig. Hypomorphic mutants of *Lk* and *Lkr* genes do not display high susceptibility to BSS.** Quantitative analyses of BSS in individual flies were performed as described in Fig 1. Data represent means ± SEM. n.s., not significant; $^*P < 0.05$ as determined by Mann-Whitney U test (seizure index, n = 60 in 3 independent experiments for *Lk*; recovery time, n = 8–9 flies for *Lk*) or by one-way ANOVA with Holm-Sidak's multiple comparisons test (seizure index, n = 54–58 in 3 independent experiments for *Lkr*; recovery time, n = 4–10 flies for *Lkr*).
(TIF)

**S13 Fig. Optogenetic silencing of LK neurons per se does not induce seizure-like behaviors in the absence of a mechanical stimulus.** Transgenic flies were crossed and kept in constant dark. Any behavioral changes upon exposure to blue (no silencing) or amber light (silencing by eNpHR) condition were examined accordingly. Data represent means ± SEM (seizure index, n = 30 flies in 3 independent experiments; recovery time, n = 2–6 flies). No significant differences in seizure index and recovery time were detected by two-way ANOVA with Holm-Sidak's multiple comparisons test.
(TIF)

**S1 Movie. A video clip for seizure-like behaviors in homozygous *Indy*²⁰⁶ mutant.**
(MP4)

**S1 Data. Numerical raw data in Figs 1, 2, 3, 4, 5, 6, and 7, S1, S2, S4, S5, S6, S7, S8, S10, S11, S12 and S13 Figs.**
(XLSX)

## Acknowledgments

We thank A. DiAntonio, S.J. Certel, D. Dickman, T. Kitamoto, D. Kuebler, R.S. Stowers, Bloomington Drosophila Stock Center, Korea Drosophila Resource Center, National Institute of Genetics, Vienna Drosophila Resource Center, Drosophila Genomics Resource Center, and Developmental Studies Hybridoma Bank for reagents; E.Y. Suh at Chungnam National University for amino acid analyses.

## Author Contributions

**Conceptualization:** Chunghun Lim.

**Formal analysis:** Jiwon Jeong, Chunghun Lim.

**Funding acquisition:** Jongbin Lee, Chunghun Lim.

**Investigation:** Jiwon Jeong, Jongbin Lee.

**Methodology:** Jiwon Jeong, Jongbin Lee, Ji-hyung Kim.

**Supervision:** Chunghun Lim.

**Validation:** Jiwon Jeong.

**Visualization:** Jiwon Jeong, Chunghun Lim.

**Writing – original draft:** Jiwon Jeong, Jongbin Lee, Chunghun Lim.

**Writing – review & editing:** Chunghun Lim.

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
