## [Decision Letter · Decision Letter 0]

27 Apr 2021

Dear Dr Lim,

Thank you very much for submitting your Research Article entitled 'Metabolic flux from the Krebs cycle to glutamate transmission tunes a neural brake on seizure onset' to PLOS Genetics.

The manuscript was fully evaluated at the editorial level and by independent peer reviewers. The reviewers appreciated the attention to an important problem, but raised some substantial concerns about the current manuscript. Based on the reviews, we will not be able to accept this version of the manuscript, but we would be willing to review a much-revised version. We cannot, of course, promise publication at that time.

If you decide to revise the manuscript for further consideration at PLOS Genetics, please aim to resubmit within the next 60 days, unless it will take extra time to address the concerns of the reviewers, in which case we would appreciate an expected resubmission date by email to plosgenetics@plos.org.

[LINK]

We are sorry that we cannot be more positive about your manuscript at this stage. Please do not hesitate to contact us if you have any concerns or questions.

Yours sincerely,

Gaiti Hasan

Associate Editor

PLOS Genetics

Gregory P. Copenhaver

Editor-in-Chief

PLOS Genetics

Reviewer's Responses to Questions

**Comments to the Authors:**

Reviewer #1: The Drosophila Indy gene encodes a fly homolog of the mammalian SLC13A5 citrate transporter. In this study, Jeong et al. found that loss-of-function Indy mutants displayed seizure-like behaviors in response to strong mechanical disturbances. Based on the experimental results obtained using genetic, optogenetic, and pharmacological approaches, the authors concluded that this behavioral abnormality of Indy mutants are caused by defective glutamate synthesis and glutamatergic transmission in the lateral horn leucokinin (LHLK) neurons, innervating the fan-shaped body. This is a carefully done study and findings are novel and interesting. In particular, it is surprising that, although Indy is expressed widely, seizure behaviors of the mutants are attributable to reduced Indy function in a specific pair of glutamatergic neurons in the brain (LHLK neurons).

Several points need clarifying and certain statements require further justification.

A number of Drosophila “bang-sensitive” mutants (i.e., mutants with increased susceptibility to seizures following mechanical stimulation) have been identified. Some of them (e.g., easily shocked, bang senseless, and slamdance) have been extensively studied. Are the “bang-induced” seizure-like behaviors observed in Indy mutants similar to or different from the known bang-sensitive phenotypes? This is the first time that Indy mutants’ bang-induced phenotype is described. Please describe the phenotype in more detail in the context of other bang-sensitive phenotypes. It would be useful if a video clip showing the representative phenotype would be included in the supplemental information or a schematic figure describing the behavioral sequence could be shown (e.g., Fig. 1 in Howlett et al. (2013) for para[bss1]).

Unlike mammals, Drosophila uses glutamate, instead of acetylcholine, as an excitatory neurotransmitter for motor neurons. The authors claim that reduced Indy function in glutamatergic neurons leads to suppression of glutamatergic transmission. It is expected that motor neurons in Indy mutants are suboptimal and that the mutants display defects in motor neuron physiology and motor behavior. Please refer to relevant previous studies, if any, that are consistent with the proposed effects of Indy loss-of-function on glutamatergic neurons.

In Indy mutants and VGlu-Gal4 mediated Indy knockdown flies, motor neurons are defective due to suppression of glutamatergic neurotransmission. In contrast, motor neurons are fully functional in the flies with Lk-Gal4 mediated Indy knockdown. When the bang sensitivity is increased due to reduced Indy function in LHLK neurons, I expect that the latter flies (Lk-Gal4 mediated Indy knockdown) display more severe seizures compared to the former flies (Indy mutants and VGlu-Gal4 mediated Indy knockdown flies) because of the difference in motor activity. However, the bang-sensitive seizure-like phenotypes are similar in these flies (Figs. 1B, 2B, and 5A). Any explanation?

Various Gal4 lines, including NP and R-lines, were used in a Gal4 mini screen (S1 Fig). Please indicate their specificity (tissue or cell type), if known, in the supporting information.

Knockdown of Indy in glutamatergic neurons leads to an increased sensitivity to bang-induced seizures. How specific is this behavioral phenotype to Indy knockdown in glutamatergic neurons? Indy knockdown in cholinergic, GABAergic, or serotonergic neurons does not have any effect? Do the NP- and R-lines used in this study include Gal4 lines driving Indy-RNAi expression in neuronal subsets other than glutamatergic neurons?

Page 2, 2nd paragraph: It’s not appropriate to call mouse SLC13A5 INDY.

Page 4, the 1st sentence: “Glutamate and GABA are two major neurotransmitters that mediate transmission at excitatory and inhibitory synapses, respectively”. In flies, they are acetylcholine and GABA, respectively.

S4B Fig: Describe how the experiment was done.

Reviewer #2: Overall recommendation: Accept with revisions

In this well-controlled and interesting manuscript, Jeong et al find that knockdown or knockout of Indy, the Drosophila homologue of a gene implicated in a human seizure disorder, causes a bang-sensitive seizure behavior in flies. They then outline a mechanism by which losing Indy function leads to decreased TCA cycle activity and associated glutamate production, which acts as the primary cellular mechanism underpinning the bang-sensitive seizure susceptibility. Screening for neurons that can recapitulate this global phenotype, they then discover a set of neurons that seem to be responsible for gating the Indy¬-associated bang-sensitive phenotype and further map aspects of a preliminary neural circuit for bang-sensitive seizures. This circuit mechanism is bolstered with both genetic knockouts/knockdowns and optogenetic manipulation. Together, then, this manuscript establishes two interesting findings: it uncovers a potential cellular mechanism for a disease-causing mutation in humans and outlines the beginnings of a novel neural circuit governing a bang-sensitive seizure phenotype in flies. The manuscript is clearly written and the experiments robust and well-controlled and will likely prove interesting to a wide audience.

Major Concerns

1). The authors use single cell data from another publication to demonstrate that there is evidence of glutamatergic signaling machinery in LK neurons. This data as presented seems to rely on a single cell in the database that expresses both LK and VGlut, which could be correctly identifying the studied neurons or could easily be due to statistical randomness. Given the fact that this claim goes against the assumptions of the field, it would be more convincing to provide staining, in situ hybridization, or tagged protein localization evidence of Vglut in these LK neurons. Additionally, more information on how the single-cell data was visualized and manipulated should be included in the methods section if this data is retained in the final manuscript.

2) The manipulation of GDH appears to this Reviewer to be an especially critical experiment as it serves as a clear link between TCA metabolism and glutamate transmission (as opposed to other downstream effects of TCA metabolism such as cell health). Therefore, I would find it extra convincing for the phenotype demonstrated using epigallocatechin gallate feeding to be recapitulated with a temperature sensitive RNAi or similar genetic approach. Alternatively, if GDH overexpression suppressed the effect of the GDH inhibitor in a cell-specific fashion this would also add certainty to that result.

3) The assumption of many experiments involving VGlut overexpression appears to be that overexpressing VGlut will compensate for lower levels of glutamate by increasing synaptic loading. Because the authors already have the capacity for GluSnFr visualization as shown later in the manuscript, it would be compelling to show that VGlut overexpression leads to greater extracellular glutamate release in the Indy-RNAi background, as such an experiment would argue against alternative changes in glutamate/glutamine cycling induced by VGlut overexpression.

4) The authors use a bang-sensitive protocol that is different from most other labs - they vortex for 25 second (long) versus 10 seconds (normal), and the "seizures" are much shorter--they observe ~20 seconds versus 1-2 minutes in bang-sensitive mutants. Is this really seizure? There is no electrophysiology to address that. In addition, the pathway they have chosen to explore and many of the manipulations would be expected to have serious consequences on metabolism (e.g. TCA cycle)...is it possible that rather than siezure this is an energy deficiency? Are there controls they could do to address this?

Minor Concerns

• The paper introduces both Indy and rogdi as Drosophila homologues of Kohlschütter-Tönz syndrome and mentions both in the introduction and discussion. However, while a small amount of compelling evidence is presented in the supplements that a similar mechanism links rogdi and indy loss-of-function to this seizure model, the vast majority of the mechanistic characterization of the manuscript focuses on indy and thus the extent of discussion of rogdi in the introduction appears somewhat speculative.

• The labels of Figure 2c are difficult to read. I fully understand the authors’ decision to only have one set of x-axis labels to save space, but the complexity of the x-axis labels makes it very difficult to understand the upper graph.

• The authors refer to depleting INDY expression – only Indy transcripts can change in expression, INDY protein should change in abundance (if this is indeed what is indicated by the change in nomenclature here)

• Several times the authors use “On the other hand” in the context of information that is logically consistent with the previous sentence, which confused me as this phrase traditionally introduces information that is contradictory to the previous sentiment in some way.

• Removal of the word “the” before metabolic challenge in the introduction.

• I would like the praise the authors for not attempting to statistically analyze the curves of seizure behavior over time in addition to seizure index and recovery time as such analysis quickly get statistically dubious.

Reviewer #3: In the manuscript titled “Metabolic flux from the Krebs cycle to glutamate transmission tunes a neural brake on seizure onset”, Jeong et al. show that mutations in the citrate transporter gene Indy can lead to bang-sensitive seizures in Drosophila. Based on extensive genetic data, they propose a model which suggest that Indy affects glutamate metabolism in a pair of glutamatergic neurons, which act as an excitatory break on the onset of more generalized seizures. This is a well-written manuscript, which provides diverse empirical evidence for the proposed model. However, several concerns about data robustness and experimental approach require additional clarifications. Below are comments that I hope the authors will find useful:

1. The mechanistic model proposed here is based on possibly interesting Drosophila data. However, its true relevance to the early-onset of seizures in KTS might be overstated. The authors should make it clear to the readers that in contrast to the mammalian brain, the primary excitatory neurotransmitter in the fly brain is acetylcholine, not glutamate. Instead, glutamate plays a key role as the primary excitatory neurotransmitter at the fly’s NMJ. Therefore, if the link between the TCA cycle and neuronal glutamate metabolism is conserved across the fly and human nervous systems then one might expect the effects of Indy mutations to lead to higher seizure threshold relative to wild type animals.

2. Some of the data presented in the supplementary figures is confusing. For example, the authors use the pan-neuronal elav-GAL4 driver to show that the Indy RNAi transgene is effective in reducing Indy mRNA levels. Yet, the elav>Indy-RNAi F1 flies show no seizure hypersensitivity. It is puzzling why the authors did not also include the well-established GAL4 drivers for GABAergic and cholinergic neurons. Did the author include UAS-Dcr transgene in all their focal crosses?

3. Linking metabolism to hyperexcitability and other aspects of neuronal physiology is always difficult because demonstrating specificity of genetic and pharmacological manipulations on phenotype, especially when it is measured at the organismal level, is almost impossible. The authors should acknowledge this in their introduction and discussion, and should address the many possible mechanistic confounds related to their proposed interpretation of the data. For example, long-term knockdown of Indy in any cell type will lead to physiological stress, and poor phenotypic outcomes

4. One major issue with the experimental design as described in the methods is that the authors may have not addressed some the known issues with strong effect of genetic background on bang-sensitivity. No specific information for most GAL4 lines used for the screen shown in Fig. S1, or what was the specific genetic background of control RNAi animals is not provided. Furthermore, the specific RNAi line used by the authors in their screen is pupal lethal when broadly expressed, and has been discarded by the NIG due to off-target effects, and no longer available.

5. If five flies were used in each trial bang-sensitivity trial, the lowest % recovery possible at any given time point is 20% (1/5 flies). Therefore, it is puzzling how it is possible to have values below 20% in Fig. 1. Furthermore, since each recovering fly represents 20% of the population, it is confusing why the curves seem to describe continuous data.

6. Previous studies have indicated that measuring seizure thresholds shifts during the refractory period between bang-induced seizures is a more robust measure of the effects of genotype on seizure sensitivity (e.g., Kuebler and Tanouye, J Neurophys 2000).

7. In my opinion, the most important, and somewhat surprising data presented here is included in the section titled “A pair of glutamatergic neurons mediates Indy-dependent seizure suppression”. However, as written, this section is very confusing, and the rational for the testing the hypothesis that Indy is particularly important in “LK” neurons, and what might be the relationship between these neuropeptidergic neurons and the glutamatergic system is not explained or justified. Furthermore, the rationale for the genetic scheme used, how the various tools used specifically address the key questions, and how the approach could help separating developmental versus physiological effects of Indy knockdown, are impossible to follow.

8. The detailed info about the ROGDI gene in the introduction is not really relevant to the main focus of the current paper.

**Have all data underlying the figures and results presented in the manuscript been provided?**

Reviewer #1: Yes

Reviewer #2: Yes

Reviewer #3: **No: **Couldn't find the raw behavioral data.

PLOS authors have the option to publish the peer review history of their article (what does this mean?). If published, this will include your full peer review and any attached files.

Reviewer #1: No

Reviewer #2: **Yes: **Marc Freeman

Reviewer #3: No

---

## [Decision Letter · Decision Letter 1]

5 Oct 2021

Dear Dr Lim,

Thank you very much for submitting your Research Article entitled 'Metabolic flux from the Krebs cycle to glutamate transmission tunes a neural brake on seizure onset' to PLOS Genetics.

The manuscript was fully evaluated at the editorial level and by independent peer reviewers. The reviewers appreciated the attention to an important topic but identified some concerns that we ask you address in a revised manuscript

We therefore ask you to modify the manuscript according to the review recommendations. Your revisions should address the specific points made by each reviewer.

[LINK]

Yours sincerely,

Gaiti Hasan

Associate Editor

PLOS Genetics

Gregory P. Copenhaver

Editor-in-Chief

PLOS Genetics

Reviewer's Responses to Questions

**Comments to the Authors:**

Reviewer #1: The authors have conducted additional experiments, provided new data, and revised the text appropriately in response to most of the comments made by reviewers. Although it is not clear why the seizure-suppressing activity of LHLK neurons are particularly sensitive to reduced glutamatergic transmission, and why their dysfunction induces seizure-like behavior, this study has set the foundation for future investigation of a novel neural circuit governing a bang-sensitive phenotype as well as potential genetic interactions between Indy and other bang-sensitive mutants.

“Seizure-like behaviors, including wing-flapping and abdomen-twitching, were scored in individual flies throughout the recorded video clip”. It’s still not clear to me exactly how seizures are defined. The fly in the video clip displays neither wing-flapping nor abdomen-twitching. Initial paralysis, loss of posture, and uncoordinated leg movements are also considered “seizure”? Please explain.

Reviewer #2: The authors have done a good job of addressing my concerns. I have no remaining concerns and think this will be a nice addition to PLoS-Genetics.

Reviewer #3: The revised manuscripts is significantly improved. One possibly minor issue that was not addressed in the revision relates to the specific Indy UAS-RNAi line used and it's possible off-target effects. This is particularly important because the transgenic line used in this study is no longer available from public sources, which potentially would make it difficult to replicate the studies by others.

**Have all data underlying the figures and results presented in the manuscript been provided?**

Reviewer #1: Yes

Reviewer #2: Yes

Reviewer #3: Yes

PLOS authors have the option to publish the peer review history of their article (what does this mean?). If published, this will include your full peer review and any attached files.

Reviewer #1: No

Reviewer #2: No

Reviewer #3: No

---

## [Editor Report · Decision Letter 2]

11 Oct 2021

Dear Dr Lim,

We are pleased to inform you that your manuscript entitled "Metabolic flux from the Krebs cycle to glutamate transmission tunes a neural brake on seizure onset" has been editorially accepted for publication in PLOS Genetics. Congratulations!

Yours sincerely,

Gaiti Hasan

Associate Editor

PLOS Genetics

Gregory P. Copenhaver

Editor-in-Chief

PLOS Genetics

Comments from the reviewers (if applicable):

**Data Deposition**

http://datadryad.org/submit?journalID=pgenetics&manu=PGENETICS-D-21-00356R2

**Press Queries**

---

## [Editor Report · Acceptance letter]

15 Oct 2021

PGENETICS-D-21-00356R2 

Metabolic flux from the Krebs cycle to glutamate transmission tunes a neural brake on seizure onset 

Dear Dr Lim, 

We are pleased to inform you that your manuscript entitled "Metabolic flux from the Krebs cycle to glutamate transmission tunes a neural brake on seizure onset" has been formally accepted for publication in PLOS Genetics! Your manuscript is now with our production department and you will be notified of the publication date in due course.

With kind regards,

Zsofia Freund

PLOS Genetics

On behalf of:
